# *FOXP2* confers oncogenic effects in prostate cancer

Xiaoquan Zhu[1]*, Chao Chen[2,3], Dong Wei[4], Yong Xu[5,6], Siying Liang[7], Wenlong Jia[8], Jian Li[1], Yanchun Qu[5], Jianpo Zhai[9], Yaoguang Zhang[4], Pengjie Wu[4], Qiang Hao[10], Linlin Zhang[11], Wei Zhang[12], Xinyu Yang[13], Lin Pan[14], Ruomei Qi[1], Yao Li[15], Feiliang Wang[16], Rui Yi[1], Ze Yang[1]*, Jianye Wang[4]*, Yanyang Zhao[1]*

[1]The Key Laboratory of Geriatrics, Beijing Hospital, National Center of Gerontology, National Health Commission, Institute of Geriatric Medicine, Chinese Academy of Medical Sciences, Beijing, China; [2]Department of Thoracic Surgery, Peking University Shenzhen Hospital, Shenzhen Peking University, Shenzhen, China; [3]The Hong Kong University of Science and Technology Medical Center, Hong Kong, China; [4]Department of Urology, Beijing Hospital, National Health Commission, Institute of Geriatric Medicine, Chinese Academy of Medical Sciences, Beijing, China; [5]Tianjin Institute of Urology, Second Hospital of Tianjin Medical University, Tianjing, China; [6]Department of Urology, Second Hospital of Tianjing Medical University, Tianjing, China; [7]Genetic Testing Center, Qingdao Women and Children's Hospital, Qingdao, China; [8]Department of Computer Science, City University of Hong Kong, Hong Kong, China; [9]Department of Urology, Beijing Jishuitan Hospital, Beijing, China; [10]Department of Urology, Beijing Tian Tan Hospital, Capital Medical University, Beijing, China; [11]School of Nursing, Harbin Medical University, Harbin, China; [12]Department of Pathology, Beijing Hospital, National Health Commission, Institute of Geriatric Medicine, Chinese Academy of Medical Sciences, Beijing, China; [13]Department of Urology, Peking University First Hospital, Institute of Urology, Beijing, China; [14]Clinical Institute of China-Japan Friendship Hospital, Beijing, China; [15]Department of Surgery, Beijing Hospital, National Health Commission, Institute of Geriatric Medicine, Chinese Academy of Medical Science, Beijing, China; [16]The Department of Ultrasonography, Beijing Hospital, National Health Commission, Institute of Geriatric Medicine, Chinese Academy of Medical Sciences, Beijing, China

*For correspondence:
zhuxiaoquan3692@bjhmoh.cn
(XZ);
yangze2806@bjhmoh.cn (ZY);
wangjy@bjhmoh.cn (JW);
zhaoyanyang3967@bjhmoh.cn
(YZ)

Competing interest: The authors declare that no competing interests exist.

**Abstract** Identification oncogenes is fundamental to revealing the molecular basis of cancer. Here, we found that *FOXP2* is overexpressed in human prostate cancer cells and prostate tumors, but its expression is absent in normal prostate epithelial cells and low in benign prostatic hyperplasia. *FOXP2* is a FOX transcription factor family member and tightly associated with vocal development. To date, little is known regarding the link of *FOXP2* to prostate cancer. We observed that high *FOXP2* expression and frequent amplification are significantly associated with high Gleason score. Ectopic expression of *FOXP2* induces malignant transformation of mouse NIH3T3 fibroblasts and human prostate epithelial cell RWPE-1. Conversely, *FOXP2* knockdown suppresses the proliferation of prostate cancer cells. Transgenic overexpression of *FOXP2* in the mouse prostate causes prostatic intraepithelial neoplasia. Overexpression of *FOXP2* aberrantly activates oncogenic MET signaling and inhibition of MET signaling effectively reverts the *FOXP2*-induced oncogenic phenotype. CUT&Tag assay identified FOXP2-binding sites located in *MET* and its associated gene *HGF*. Additionally, the novel recurrent *FOXP2-CPED1* fusion identified in prostate tumors results in high expression of truncated FOXP2, which exhibit a similar capacity for malignant transformation. Together, our data indicate that *FOXP2* is involved in tumorigenicity of prostate.

## Editor's evaluation

The authors convincingly showed FOXP2 expression being associated with a high Gleason score, and ectopic expression of FOXP2 inducing malignant transformation of non-tumor cells. This is associated with increased MET signalling. With this, the authors position FOXP2 as bona-fide oncogene, driving prostate cancer development.

## Introduction

Oncogenes arise as a consequence of genetic alterations that increase the expression level or activity of a proto-oncogene. Activation of oncogenes causes malignant transformation of normal cells and maintains cancerous cell proliferation and survival. Clinical intervention strategies targeting oncoproteins and their related signaling pathways lead to growth arrest and apoptosis in cancer cells (*Weinstein and Joe, 2008*; *Luo et al., 2009*).

Prostate cancer is the second most commonly diagnosed cancer worldwide in men, with ~1.3 million new cases and ~0.36 million deaths in 2018 (*Culp et al., 2020*). In China, the incidence rate of prostate cancer has increased quickly, with an annual percentage change of ~13% since 2000 (*Chen et al., 2016*). These findings stress the importance of a comprehensive understanding of the molecular mechanisms underlying the genesis of prostate cancer. Accumulating data from genomic characterization and functional studies have revealed some oncogenes, including the E26 transformation-specific (ETS) family genes *ERG* and *ETV1*, among others, whose overexpression is involved in prostate tumorigenesis (*Baena et al., 2013*; *Carver et al., 2009*; *Tomlins et al., 2005*). However, the oncogenes that contribute to the initiation of the disease remain unclear.

In this study, we identified a previously unknown intrachromosomal gene fusion involving *FOXP2* in primary prostate tumors by RNA-Seq and RT-PCR combined with Sanger sequencing. The fusion encodes a truncated FOXP2 mutant protein encompassing the key domains and is highly expressed in the fusion-carrying prostate tumor. *FOXP2* encodes a highly conserved forkhead box transcription factor and is tightly associated with vocal development in humans and mammals (*Bowers et al., 2013*; *Enard et al., 2009*; *Fisher and Scharff, 2009*). Mutations in *FOXP2* are the only known cause of developmental speech and language disorder in humans (*Spiteri et al., 2007*). Most of the recent studies investigating the roles of *FOXP2* in cancer have focused on noncoding RNA-mediated dysregulation of *FOXP2*, suggesting that *FOXP2* has either oncogenic or tumor-suppressive effects on cancer development in different tumor contexts (*Cuiffo et al., 2014*; *Du et al., 2020*; *Kim et al., 2019*; *Xu et al., 2022*; *Zhao et al., 2021*). However, to date, little is known regarding the link between *FOXP2* and prostate cancer. Here, we investigated the expression of *FOXP2* in prostate cancer samples. Moreover, we explored the biological functions of *FOXP2* in human prostate tumors and prostate cancer cell lines. In addition, the impact of prostate-specific expression of the *FOXP2* gene in mice was tested.

## Results

### Enhanced *FOXP2* expression in prostate cancer plays an oncogenic role

*FOXP2* mutations have long been thought to be the cause of developmental speech and language disorder in humans (*Enard et al., 2009*; *Fisher and Scharff, 2009*; *Spiteri et al., 2007*). In this study, we identified a new gene fusion, *FOXP2-CPED1*, in 2 of 100 indolent prostate tumors by performing RNA sequencing and whole-genome sequencing analyzes (*Figure 1A and B*, *Figure 1—figure supplement 1A–E*, and *Supplementary file 1a and b*). We found that in the prostate tumor the fusion consequently resulted in increased expression of a truncated FOXP2 protein that retained the complete FOXP2 functional domains (*Spiteri et al., 2007*; *Hannenhalli and Kaestner, 2009*; *Lai et al., 2001*; *Shu et al., 2001*), but had an aberrant C-terminus (*Figure 1C and D*, *Figure 1—figure supplement 1F and G*). Mechanistically, we conducted small RNA sequencing of the fusion-positive tumor followed by functional assays, demonstrating that *FOXP2-CPED1* fusion led to loss of the 3′UTR of *FOXP2*, thus allowing escape from regulation by miR-27a and miR-27b and consequently resulting in aberrantly high expression of the truncated FOXP2 protein with full functional domains (*Figure 1—figure supplement 1H–L*).

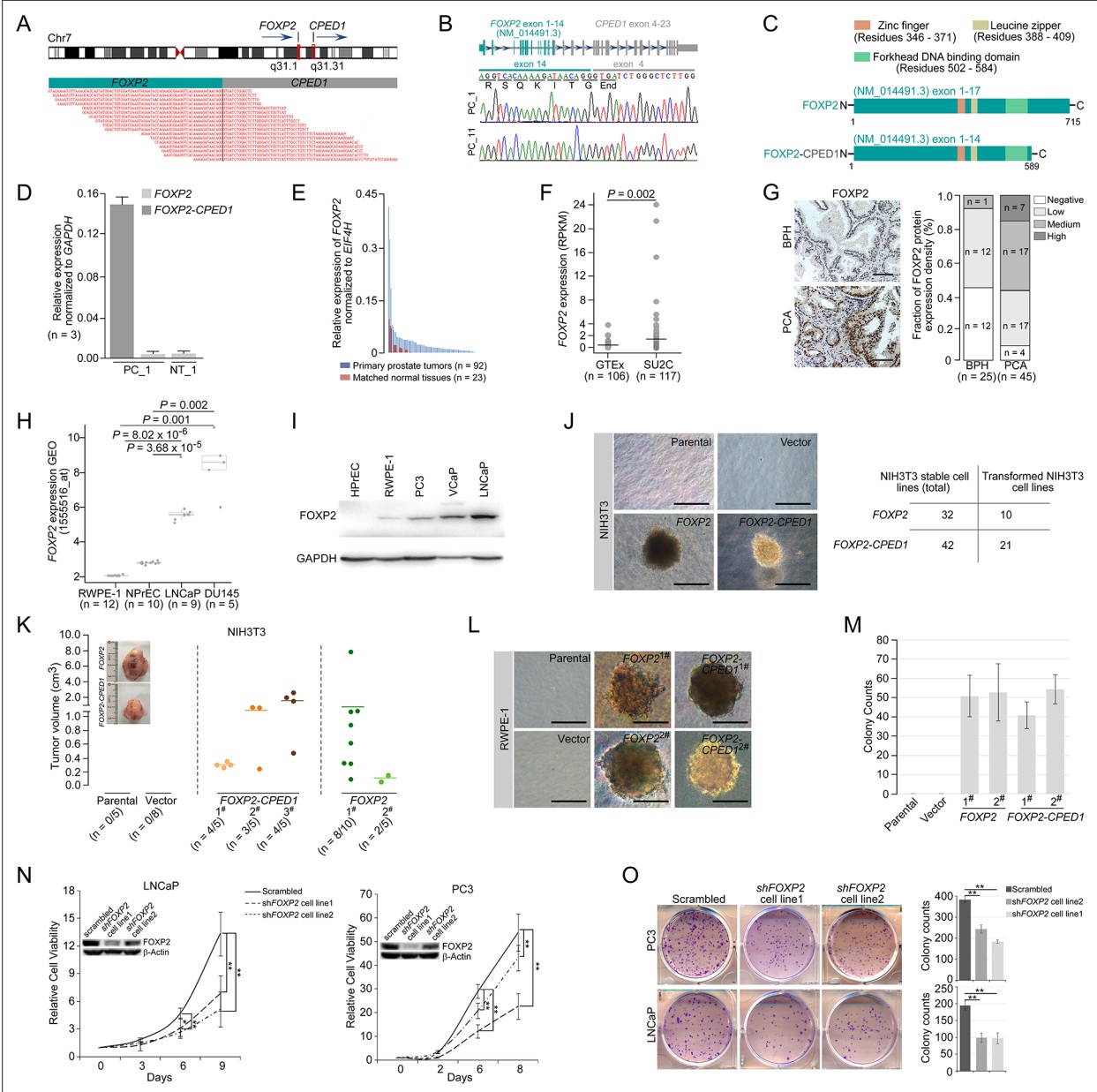

**Figure 1.** Oncogenic roles of *FOXP2* in prostate cancer. (**A**) *FOXP2-CPED1* fusion gene identified in the prostate tumor of patient (PC_1) by RNA-seq. Top: schematic of chromosome 7 with the position and strand orientation indicated for the *FOXP2* and *CPED1* gene. Bottom: schematic representation of the paired-end reads covering the junction between *FOXP2* and *CPED1*. (**B**) *FOXP2-CPED1*-specific PCR from cDNA derived from the prostate tumors (PC_1 and PC_11). Sanger sequencing chromatogram of the PCR products showing the reading frame encompassing the breakpoint in the two fusion-positive tumors (PC_1 and PC_11). The fusion transcript sequence was identical in the two cases at the fusion junction (exon 14 of *FOXP2* fused to exon 4 of *CPED1*), resulting in a stop-gain mutation right after the fusion breakpoint. (**C**) Schematic of protein structures of wild-type FOXP2 and truncated FOXP2 encoded by the *FOXP2-CPED1* fusion transcript. FOXP2-CPED1 fusion protein has an aberrant C-terminus but it retains the complete FOXP2 functional domains. (**D**) Expression of natural *FOXP2* and *FOXP2-CPED1* fusion in the tumor (PC_1) and its matched normal tissue (NT_1) by qPCR. Mean ± SD. (**E**) Expression of *FOXP2* mRNA in 92 primary prostatic adenocarcinoma tissues and 23 matched normal tissues analyzed by qPCR. (**F**) Expression of *FOXP2* mRNA in normal prostate samples from the GTEx dataset (n = 106) and metastatic prostate tumors from the SU2C dataset (n = 117). The horizontal bar indicates the mean in each group. p-Values were calculated by two-tailed Mann–Whitney *U*-test. (**G**) Left: representative images of FOXP2 protein expression in benign prostatic hyperplasia (BPH) (n = 25) or primary prostatic adenocarcinoma (PCA) (n = 45) by immunohistochemistry. Scale bars, 100 μm. *Right*: the bar chart indicates the number of PCA and BPH with negative, low, medium, and high FOXP2 expression, respectively. (**H**) Expression of *FOXP2* mRNA in two types of human benign prostate epithelial cells, RWPE-1 and NPrEC, and in two prostate cancer cell lines LNCaP and DU145 from the GEO database (1555516_at). p-Values were calculated by two-tailed Student's *t*-test. (**I**) Western blot measuring FOXP2 protein in one normal human prostate epithelial cell HPrEC and one benign prostate epithelial cell RWPE-1, and in three human prostate cancer cell lines, PC3, VCaP,

*Figure 1 continued on next page*

*Figure 1 continued*

and LNCaP. The experiment was repeated twice with similar results. (**J**) Cell transformation and anchorage-independent growth were measured by a soft agar assay in parental NIH3T3 cells and NIH3T3 cells stably expressing the empty vector, *FOXP2* and *FOXP2-CPED1*. Repeated 4-week assays were performed. Scale bars, 100 µm. See *Figure 1—figure supplement 3D* for details. Table indicating the number of total NIH3T3 cell lines overexpressing *FOXP2* (n = 32) or *FOXP2-CPED1* (n = 42) tested by soft agar colony formation assay (left column) and the number of transformed NIH3T3 cell lines induced by *FOXP2* (n = 10) or *FOXP2-CPED1* (n = 21) (right column). See *Figure 1—figure supplement 3D* for details. (**K**) Quantification of the tumor volume in NOD-SCID mice injected with NIH3T3 cells stably expressing vector, *FOXP2* (two cell lines) or *FOXP2-CPED1* (three cell lines) for 8 wk. Inset: representative images of xenografted tumors derived from the corresponding cells. See *Figure 1—figure supplement 3E and F* for details. (**L, M**). (**L**) Soft agar assay showing that RWPE-1 cells stably expressing *FOXP2* (two cell lines) or *FOXP2-CPED1* (two cell lines) were able to form colonies. Scale bars, 100 µm; (**M**) Quantitative analysis of the colony formation ability of the corresponding RWPE-1 cells. Mean ± SD; n = 4. (**N**) Cell growth of prostate cancer cell lines (LNCaP and PC3) stably expressing control vector or *FOXP2* shRNA (two clones, #1 and #2). *$p<0.05$, **$p<0.005$ by two-tailed Student's *t*-test, mean ± SD; n = 4. Inset: immunoblot showing knockdown of FOXP2 protein in LNCaP or PC3 cells. (**O**) Left: focus formation assay performed with PC3 or LNCaP cells stably expressing control vector or *FOXP2* shRNA (two clones, #1 and #2). Right: bar graph showing the number of colonies formed by PC3 or LNCaP cells after *FOXP2* silencing. **$p<0.005$ by two-tailed Student's *t*-test; mean ± SD; n = 4.

The online version of this article includes the following source data and figure supplement(s) for figure 1:

**Source data 1.** Uncropped blot for *Figure 1I*.

**Source data 2.** Uncropped blot for *Figure 1N*.

**Figure supplement 1.** Identification of *FOXP2-CPED1* fusion gene in human prostate tumors.

**Figure supplement 1—source data 1.** Uncropped gel for *Figure 1—figure supplement 1A*.

**Figure supplement 1—source data 2.** Uncropped gel for *Figure 1—figure supplement 1B*.

**Figure supplement 1—source data 3.** Uncropped blot for *Figure 1—figure supplement 1G*.

**Figure supplement 1—source data 4.** Uncropped blot for *Figure 1—figure supplement 1K*.

**Figure supplement 1—source data 5.** Uncropped blot for *Figure 1—figure supplement 1L*.

**Figure supplement 2.** Oncogenic roles of *FOXP2* in prostate cancer.

**Figure supplement 2—source data 1.** Uncropped blot for *Figure 1—figure supplement 2A*.

**Figure supplement 3.** Oncogenic roles of *FOXP2* in prostate cancer.

**Figure supplement 3—source data 1.** Uncropped blot for *Figure 1—figure supplement 3B*.

**Figure supplement 4.** Clinical significance of *FOXP2* expression and its copy number alterations (CNAs) in prostate cancer.

We therefore evaluated the expression of *FOXP2* in prostate cancer by analyzing several datasets, including our in-house samples (primary prostate cancer tumors, n = 92; matched normal tissues, n = 23), the SU2C dataset (metastatic prostate cancer samples, n = 117) (*Robinson et al., 2015*) and the GTEx dataset (normal prostate tissues, n = 106). The *FOXP2* mRNA levels were significantly increased in prostate cancer samples with respect to normal tissues (*Figure 1E and F*). We then examined tissue extracts by immunoblotting. The amounts of FOXP2 protein were evidently increased in primary prostate adenocarcinomas relative to the matched normal tissues and benign prostatic hyperplasia (BPH) samples (*Figure 1—figure supplement 2A and B*). We further carried out an immunohistochemical assay on 25 BPH samples and 45 primary prostate tumors. Strong or medium staining of FOXP2 protein was observed in 53% (24/45) of primary prostate adenocarcinomas. No or low nuclear staining of FOXP2 was observed in 96% (24/25) of benign prostatic tissues. There was a statistically significant difference in the FOXP2 staining intensity between prostate cancer and benign prostate tissue (two-tailed $p=0.001$, by Fisher's exact test) (*Figure 1G*). Likewise, *FOXP2* expression was markedly elevated in human prostate cancer cell lines (PC3, LNCaP, VCaP, and DU145), but was undetectable or low in normal or immortalized human prostate epithelial cells (HPrEC and RWPE-1) (*Figure 1H and I*). Given the data above, these findings suggested that there is a tendency for increased expression of FOXP2 protein between normal tissue and prostate neoplasia. Together, our data indicated that *FOXP2* was highly expressed in human prostate tumors.

Overexpression of specific genes such as *ERG* (*Carver et al., 2009*; *Tomlins et al., 2007*) and *EZH2* (*Varambally et al., 2002*) results in a phenotype of hyperplasia and promotes of invasive properties in the prostate, indicating a critical proto-oncogenic role in prostate tumorigenesis. Therefore, in this study, to explore the biological consequence of high expression of the *FOXP2* gene, we first introduced the wild-type or fusion *FOXP2* cDNA into mouse NIH3T3 fibroblasts that lack endogenous Foxp2 protein expression (*Figure 1—figure supplement 3A and B*) and then carried out a colony

formation assay. NIH3T3 fibroblasts, as normal fibroblasts, are considered to be entirely anchorage-dependent for proliferation. We found that *FOXP2* can transform NIH3T3 fibroblasts, inducing loss of contact inhibition and gain of anchorage-independent growth, which are tumorigenic properties (*Figure 1J*, *Figure 1—figure supplement 3C and D*). We also noted that, as anticipated from the structural properties of the *FOXP2-CPED1* fusion, it was able to confer this oncogenic phenotype as well (*Figure 1J*, *Figure 1—figure supplement 3C and D*). The assays showed that 21 of the 42 NIH3T3 cell lines expressing the *FOXP2-CPED1* fusion gained the ability for anchorage-independent growth, as did 10 of the 32 cell lines with *FOXP2* expression, indicating that *FOXP2-CPED1* and *FOXP2* have comparable effects on cell transformation (p=0.23, by Fisher's exact test). We next determined the oncogenic activity of *FOXP2* and its rearrangement lesion in a NOD-SCID mouse model. We observed that *FOXP2*-overexpressing NIH3T3 cells injected into the flanks of NOD-SCID mice formed tumors at a higher penetrance than did parental or empty vector control cells: in 10/15 mice compared with 0/5 mice for parental cells and 0/8 for control cells (*Figure 1K*, *Figure 1—figure supplement 3E and F*). In 11/15 mice, *FOXP2-CPED1*-overexpressing NIH3T3 cells formed tumors in mice (*Figure 1K*, *Figure 1—figure supplement 3E and F*).

The oncogenic properties of *FOXP2* were further shown by its ability to transform a human prostate epithelial cell line, RWPE-1, which is immortalized but nontransformed (*Bello et al., 1997*). We observed that *FOXP2* overexpression significantly increased the proliferation of RWPE-1 cells relative to that of control cells (*Figure 1—figure supplement 3G*). Similar to NIH3T3 cells, assays of focus formation and anchorage-independent growth consistently showed that overexpression of wild-type *FOXP2* or *FOXP2* fusion induced malignant transformation of RWPE-1 cells (*Figure 1L and M*, *Figure 1—figure supplement 3H*). Conversely, we carried out *FOXP2* short hairpin RNA (shRNA) knockdown in two prostate cancer cell lines (LNCaP and PC3). We found that shRNA-mediated *FOXP2* knockdown markedly inhibited the cell growth and colony-forming ability of LNCaP and PC3 cells (*Figure 1N and O*). Our findings were consistent with previous data showing that the cell growth ability was decreased by *Foxp2* knockdown in mouse pancreatic cancer cells (*Rad et al., 2015*). Taken together, our data demonstrated that *FOXP2* is tumorigenic in prostate cancer.

## *FOXP2* overexpression aberrantly activates oncogenic MET signaling

To explore the molecular mechanism underlying the role of *FOXP2* in prostate cancer, we first carried out analysis of the entire expression spectrum of 255 primary prostate tumors from TCGA (*Figure 2A*). We observed that the expression of *FOXP2* was significantly correlated with the expression of 3206 genes (refer to *FOXP2* expression-correlated genes, FECGs) (|Spearman's $\rho \geq 0.5$|) (*Figure 2B*, *Supplementary file 2*). Gene set enrichment analysis (GSEA) of these FECGs identified that the top significantly enriched gene sets corresponded to 11 known prostate cancer gene sets, including Wallace_Prostate_Cancer_Race_Up, Acevedo_FGFR1_Targets_In_Prostate_Cancer_Model, Kondo_Prostate_Cancer_with_H3k27me3, and Kras.Prostate_Up.V1_DN signatures (*Figure 2C*, *Supplementary file 1c*). Pathway enrichment analysis identified that Dairkee_Tert_Targets_Up, PI3K-AKT Signaling, Mtiotic_Spindle, TGF_Beta_Signaling, and Inflammatory_Response were significantly enriched in positively *FOXP2*-correlated genes. E2F_Targets, DNA_Repair, and Oxidative_Phosphorylation were significantly enriched in negatively correlated genes (*Figure 2D*, *Supplementary file 1d*). Notably, 18 of the 74 FECGs enriched in the PI3K-AKT pathway are known cancer driver genes, which include 4 core members of oncogenic MET signaling (*HGF*, *MET*, *PIK3R1*, and *PIK3CA*) (*Figure 2D*).

We further examined the correlation between the MET pathway and *FOXP2* in three additional datasets including our primary prostate cancer data, GSE54460 (*Long et al., 2014*) and Taylor (*Taylor et al., 2010*). Consistently, we observed a significantly positive correlation between the expression of *FOXP2* and the four individual core members of MET signaling (*HGF*, *MET*, *PIK3R1*, and *PIK3CA*) across the three datasets (*Figure 2E*). Previous studies have reported that the receptor tyrosine kinase MET and its ligand HGF are important for the growth and survival of several tumor types, including prostate cancer (*Bradley et al., 2017*; *Humphrey et al., 1995*). These four putative candidates were further confirmed to be upregulated in RWPE-1 cells overexpressing *FOXP2* by qPCR (*Figure 2F*). A similar effect was also observed in NIH3T3 cell lines that ectopically expressed *FOXP2* (*Figure 2—figure supplement 1A–E*). Conversely, the core members (*HGF*, *MET*, and *PIK3CA*) were downregulated in the *FOXP2* knockdown PC3 prostate cancer cells (*Figure 2—figure supplement 1F*). These data strongly suggested that oncogenic MET signaing is activated by *FOXP2* in prostate tumors.

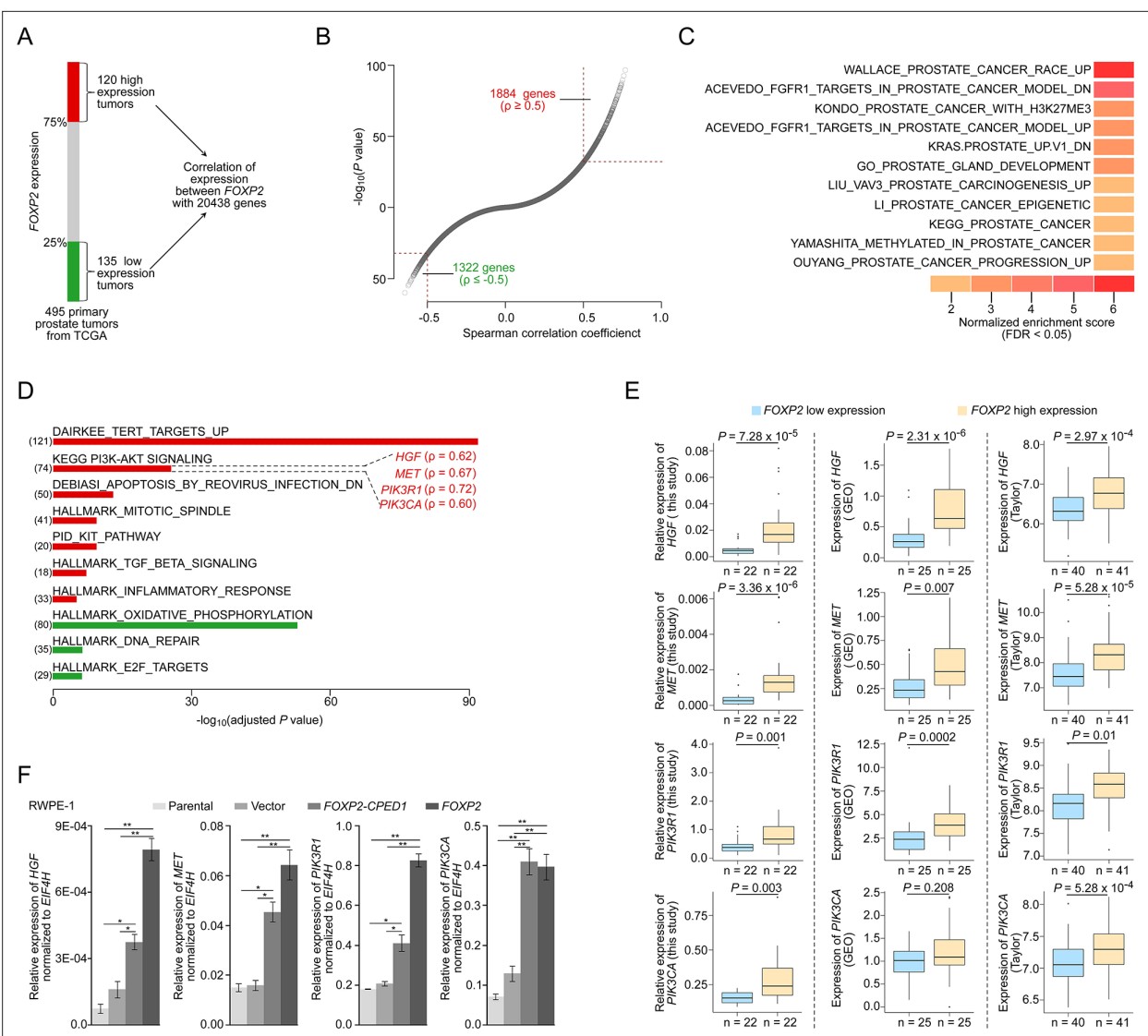

**Figure 2.** *FOXP2* activates oncogenic MET signaling in *FOXP2*-overexpressing cells and prostate tumors. (**A**) Schematic indicating the analysis of the TCGA dataset (Prostate Adenocarcinoma, Provisional, n = 495). In total, n = 255, top 25%, high expression (n = 120), bottom 25%, low expression (n = 135). (**B**) In the 255 primary prostate tumors shown in (**A**), the expression of 3206 genes was significantly correlated with the expression of *FOXP2* (|Spearman's $\rho \geq 0.5$|; 1884 significantly positively correlated genes; 1322 significantly negatively correlated genes). See also *Supplementary file 2*. (**C**) The significant enrichment of the 3206 *FOXP2* expression-correlated genes (FECGs) in known prostate cancer gene sets from the Molecular Signatures database by gene set enrichment analysis (GSEA). p-Values were calculated by two-tailed Fisher's exact test. See also *Supplementary file 1c*. (**D**) Analysis of the biological pathways of 3206 *FOXP2* expression-correlated genes using gene sets from MSigDB and KEGG. The number in parentheses corresponds to *FOXP2* expression-correlated genes that are enriched in the corresponding pathway. The Spearman correlation coefficients between the expression of *FOXP2* and the expression of four core components of MET signaling (*HGF*, *MET*, *PIK3R1*, and *PIK3CA*) are indicated in parentheses. Two-tailed p-values by Fisher's exact test and adjusted by Bonferroni correlation. See also *Supplementary file 1d*. (**E**) The correlation analysis of gene expression between *FOXP2* and MET signalling members in our primary prostate tumors (n = 92) and two other human primary prostate cancer datasets (GSE54460, n = 100; Taylor, n = 162) by qPCR. *HGF*, *MET*, *PIK3R1*, and *PIK3CA* expression levels (normalized to *EIF4H*) classified by *FOXP2* expression level (bottom 25%, low expression; top 25%, high expression). p-Values were calculated by two-tailed Mann–Whitney *U*-test. (**F**) In *FOXP2*- or *FOXP2*-*CPED1*-transformed RWPE-1 cells and control cells (parental cells and empty vector-expressing cells), the relative mRNA expression levels of *HGF*, *MET*, *PIK3R1*, and *PIK3CA* (normalized to *EIF4H*) were examined by qPCR. p-Values calculated by two-tailed Student's *t*-test, mean ± SD; n = 3. *p<0.05, **p<0.005.

The online version of this article includes the following figure supplement(s) for figure 2:

**Figure supplement 1.** *FOXP2* activates oncogenic MET signaling in *FOXP2*-overexpressing cells.

## Targeting MET signaling inhibits *FOXP2*-induced oncogenic effects

We next assessed whether activation of MET signaing plays a crucial role in *FOXP2*-driven oncogenic effects. We observed obviously elevated phosphorylation of tyrosines Y1234/1235 in the MET kinase domain in the *FOXP2*-transformed RWPE-1 and *FOXP2*-transformed NIH3T3 cells (*Figure 3A*). Furthermore, overexpression of *FOXP2* induced strong activation of PI3K signaing, as indicated by the elevated phosphorylation level of AKT at serine 473 (*Figure 3A*). AKT is a key downstream effector of HGF/MET/PI3K signaling (*Comoglio et al., 2018*) and is able to initiate prostate neoplasia in mice (*Majumder et al., 2003*). Conversely, FOXP2 silencing using shRNA caused decreases in phospho-MET and phospho-AKT levels in two prostate cancer cell lines, PC3 and LNCaP (*Figure 3B*). Moreover, we detected higher phosphorylation levels of MET and AKT in primary human prostate tumors with higher compared with lower FOXP2 protein abundance (*Figure 3C*).

Since the *FOXP2*-overexpressing cells exhibited an increase in p-MET/p-AKT levels, we hypothesized that MET and AKT inhibitors could have a therapeutic benefit. The *FOXP2*-overexpressing cells were subsequently treated with the MET tyrosine kinase inhibitor foretinib or AKT inhibitor MK2206, resulting in twofold to threefold (for foretinib) and threefold to twelvefold (for MK2206) decreases in the half-maximal inhibitory concentration ($IC_{50}$) in RWPE-1 cells and NIH3T3 cells with *FOXP2* overexpression (*Figure 3D and E*). Furthermore, treatment of *FOXP2*-overexpressing RWPE-1 cells and *FOXP2*-overexpressing NIH3T3 cells with foretinib or MK2206 abrogated MET and/or AKT phosphorylation (*Figure 3F and G*) and resulted in significantly reduced anchorage-independent growth or foci formation relative to those in nontreated cells (*Figure 3I and H*). Next, we knocked down *MET* expression in RWPE-1 cells that overexpressed the *FOXP2* or *FOXP2-CPED1*. We observed a decrease in both MET expression and phospho-AKT levels in these cells. Consistently, we found that siRNA-mediated knockdown of MET significantly reduced the growth of RWPE-1 cells (*Figure 3—figure supplement 1A*). We also conducted an experiment to rescue the expression of HA-tagged MET in *FOXP2* knockdown PC3 cells and performed a cell proliferation assay. However, the restoration of MET does not reverse the change in viability of *FOXP2* knockdown PC3 cells (*Figure 3—figure supplement 1B*). Additionally, to test whether *MET* and its associated genes are bound by FOXP2, we carried out Cleavage Under Targets and Tagmentation (CUT&Tag) assay (*Kaya-Okur et al., 2019*) in LNCaP cells overexpressing HA-tagged *FOXP2*. We identified potential FOXP2-binding fragments located in *MET* and *HGF* in prostate cancer LNCaP cells (*Figure 3J and K*, *Figure 3—figure supplement 1C and D*). Together, these results demonstrated the involvement of MET signaing in the cellular transformation driven by *FOXP2*.

## Prostate-specific overexpression of *FOXP2* causes prostatic intraepithelial neoplasia (PIN)

To further determine the effects of *FOXP2* in vivo, we generated mice with prostate-specific expression of *FOXP2* and the *FOXP2-CPED1* fusion under the control of a modified probasin promoter regulated by androgen (*Zhang et al., 2000*; *Figure 4—figure supplement 1A–E*). By 46–65 wk of age, all ARR2PB-*FOXP2* mice (n = 35) and ARR2PB-*FOXP2-CPED1* mice (n = 32) examined had hyperplasia in prostates (*Supplementary file 1e*). Furthermore, 97% (34/35) of *FOXP2* transgenic mice and 88% (28/32) of *FOXP2-CPED1* mice developed PIN. All littermate wild-type prostates had normal morphology, although they also had focal benign hyperplasia. Most PINs were observed in the anterior and ventral prostate lobes, and they were all focal (*Supplementary file 1e*). The foci of the lesion had multiple layers of atypical epithelial cells and presented papillary, tufting or cribriform patterns. The atypical cells exhibited poorly oriented and markedly enlarged nuclei with severe pleomorphism, hyperchromasia, and prominent nucleoli (*Figure 4A*, *Figure 4—figure supplement 1F*). In comparison with wild-type control mice, both transgenic *FOXP2* and *FOXP2-CPED1* fusion mice showed a five- to sixfold increase in the proliferation index of prostate epithelial cells in preneoplastic prostate glands, as shown by Ki67 staining (*Figure 4B*). Moreover, consistent with our in vitro data, increased phosphorylation levels of mouse Met and its downstream mediator Akt in prostates of both *FOXP2* and *FOXP2-CPED1* fusion transgenic mice compared with those in prostates of control mice were observed (*Figure 4C*, *Figure 4—figure supplement 1G*). Together, our data indicated that *FOXP2* has an oncogenic role in prostate tumorigenesis.

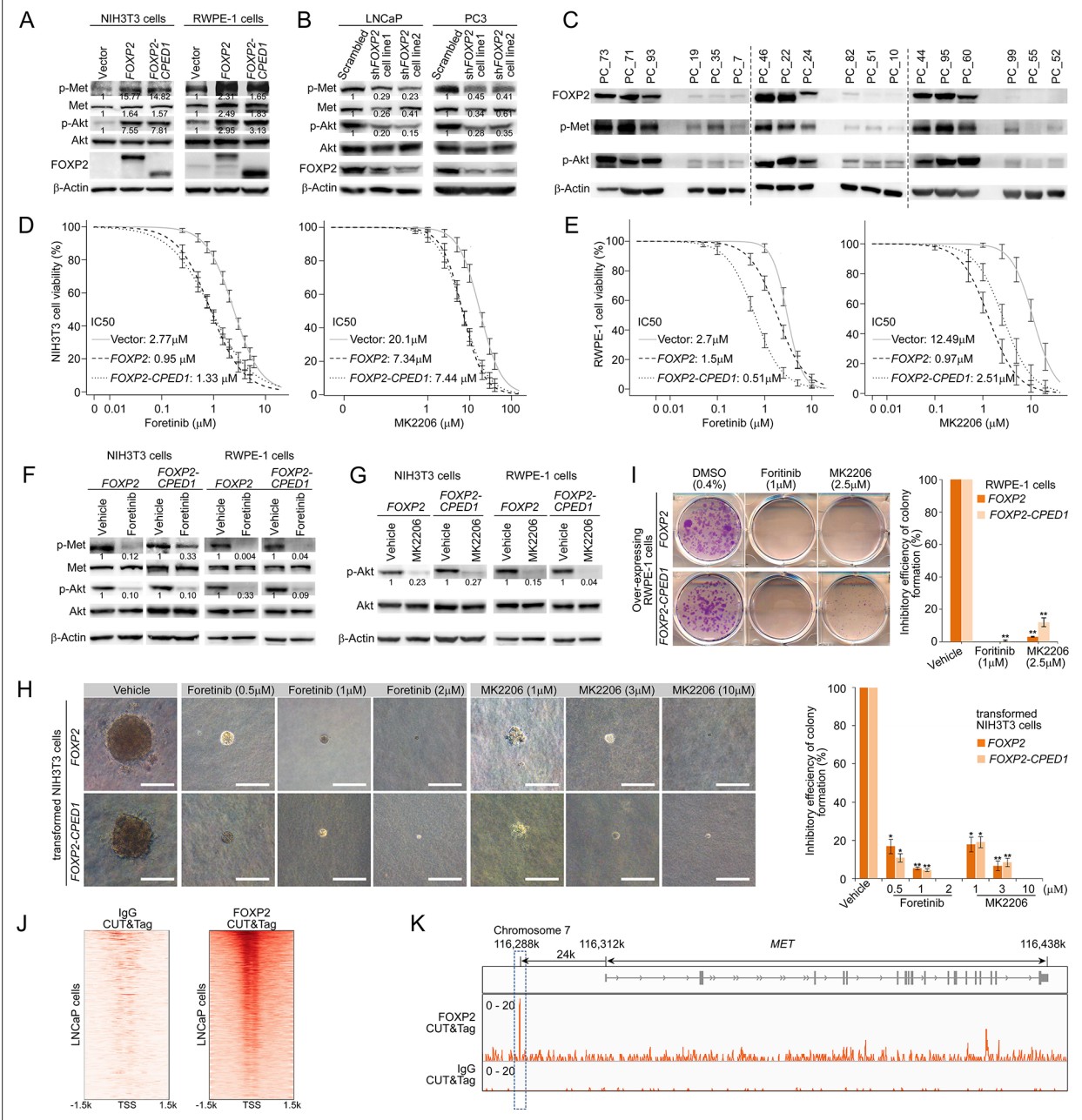

**Figure 3.** Targeting MET signaling inhibits *FOXP2*-induced oncogenic effects. (**A**) Immunoblotting of the expression levels of p-Met (Y1234/1235) and p-Akt (S473) in NIH3T3 cells overexpressing *FOXP2* or *FOXP2-CPED1* (left) or in RWPE-1 cells overexpressing *FOXP2* or *FOXP2-CPED1* (right). The experiment was repeated three times with similar results. Relative ratios of the intensities of the p-Met, p-Akt, Met, and Akt protein bands relative to the beta-Actin band are shown bottom. (**B**) Protein blot analysis of the expression of p-Met (Y1234/1235) and p-Akt (S473) in LNCaP (left) or PC3 cells (right) that stably expressed scrambled vector and *FOXP2* shRNA, respectively. The experiment was repeated three times with similar results. Relative ratios of the intensities of the p-Met, p-Akt, Met, and Akt protein bands relative to the beta-Actin band are shown bottom. (**C**) Protein blot analysis of the activity of MET (Y1234/1235) and AKT (S473) in human localized primary prostate tumors (n = 18) with high and low FOXP2 expression, respectively. (**D, E**) IC$_{50}$ curves of the vector, *FOXP2*- or *FOXP2-CPED1*-overexpressing NIH3T3 cells (**D**) or -overexpressing RWPE-1 cells (**E**) treated with the MET inhibitor foretinib or the AKT inhibitor MK2206 assessed at 48 hr after the treatment. Mean ± SD; n = 3. (**F–G**) Western blot analysis of the expression levels of p-Met (Y1234/1235) and p-Akt (S473) in the *FOXP2*- or *FOXP2-CPED1*-overexpressing NIH3T3 cells or -overexpressing RWPE-1 cells after 48 hr treatment with vehicle (0.4% DMSO) and the MET inhibitor foretinib (0.5 μM for NIH3T3 cells, 1 μM for RWPE1 cells) (**F**) or the AKT inhibitor MK2206 (1 μM for NIH3T3 cells, 2.5 μM for RWPE1 cells) (**G**). The experiment was repeated twice with similar results. Relative ratios of the intensities of the p-Met, p-Akt, and Met protein bands relative to the beta-Actin band are shown bottom. (**H**) Left: soft agar assay of the stable *FOXP2*- or *FOXP2-CPED1*-overexpressing NIH3T3 cells treated with vehicle, MET inhibitor, or AKT inhibitor. Images of representative cells treated with the MET inhibitor foretinib

*Figure 3 continued on next page*

*Figure 3 continued*

and AKT inhibitor MK2206. Scale bars, 100 µm. Right: bar graph showing the percentage of the colonies formed by these NIH3T3 cells after treatment with the indicated MET or AKT inhibitors normalized to those of DMSO-treated cells. *p<0.01, **p<0.005 by two-tailed Student's *t*-test, mean ± SD; n = 4. (**I**) Left: focus formation assay of the stable *FOXP2*- or *FOXP2-CPED1*-overexpressing RWPE-1 cells treated with vehicle, the MET inhibitor foretinib or the AKT inhibitor MK2206. Right: bar graph showing the percentage of the colonies formed by these REPW-1 cells after treatment with the indicated MET or AKT inhibitors normalized to those of DMSO-treated cells. **p<0.005 by two-tailed Student's *t*-test, mean ± SD; n = 4. (**J**) CUT&Tag density heat maps for FOXP2-occupied genomic locations within ± 1.5 kb of the transcription start site (TSS) in LNCaP cells overexpressing HA-tagged FOXP2. All peaks were rank ordered by FOXP2 distribution from high to low. (**K**) CUT&Tag identified binding site of FOXP2 in *MET*. Representative track of FOXP2 at the *MET* gene locus is shown. The blue square denotes the binding site unique to FOXP2.

The online version of this article includes the following source data and figure supplement(s) for figure 3:

**Source data 1.** Uncropped blot for *Figure 3A*.

**Source data 2.** Uncropped blot for *Figure 3B*.

**Source data 3.** Uncropped blot for *Figure 3C*.

**Source data 4.** Uncropped blot for *Figure 3F*.

**Source data 5.** Uncropped blot for *Figure 3G*.

**Figure supplement 1.** *FOXP2* activates oncogenic MET signaling in *FOXP2*-overexpressing cells.

**Figure supplement 1—source data 1.** Uncropped blot for *Figure 3—figure supplement 1A*.

**Figure supplement 1—source data 2.** Uncropped blot for *Figure 3—figure supplement 1B*.

**Figure supplement 1—source data 3.** Uncropped gel for *Figure 3—figure supplement 1D*.

## Discussion

In this study, we determined for the first time that the overexpressed FOXP2 protein causes malignant transformation of normal prostate epithelial cells in humans and mice. *FOXP2* encodes an evolutionally conserved forkhead box transcription factor and is highly expressed in human thyroid, lung, and smooth muscle tissue; it is particularly highly expressed in the brain. However, the *FOXP2* expression pattern in normal prostate tissue and malignant neoplasms of the prostate has not been clearly characterized. Our findings revealed that FOXP2 protein expression is absent or low in normal human prostate epithelial cells and benign prostatic hyperplasia but is markedly increased in PIN, localized prostate tumors, and metastatic prostate cancer cell lines. We found that primary prostate tumors with *FOXP2* expression had a higher Gleason grade than cases without *FOXP2* expression by analyzing the clinical data of 491 human TCGA primary prostate tumors (*Figure 1—figure supplement 4A*). Next, we observed frequent amplification of the *FOXP2* gene, which occurred in 15–20% of primary prostate tumors from the Broad/Cornell (*Baca et al., 2013*) and TCGA datasets and in 25% of metastatic tumors from the SU2C dataset (*Robinson et al., 2015*; *Figure 1—figure supplement 4B*). Moreover, we conducted the analysis on tumors with both *FOXP2* gene expression and amplification data available in the Catalogue Of Somatic Mutations In Cancer (COSMIC) datasets. In 268 tumors with *FOXP2* amplification at the DNA level, we observed consistent *FOXP2* mRNA overexpression across most tumors (197/268) (*Figure 1—figure supplement 4C*). Frequent *FOXP2* gain also occurred in 18 other types of human solid tumors (*Figure 1—figure supplement 4D*). Finally, we evaluated the clinical significance of *FOXP2* copy number alterations (CNAs) in 487 primary prostate tumors. CNAs of *FOXP2* were significantly associated with high Gleason scores (Gleason score ≥ 8) (OR = 2.10; 95% CI, 1.38–3.22; p=0.001, by Fisher's exact test) and high-grade pathologic T stages (T ≥ 3a) (OR = 2.01; 95% CI, 1.26–3.21; p=0.003, by Fisher's exact test) (*Figure 1—figure supplement 4E*). These data suggested that the genomic lesion in *FOXP2* might contribute to high-risk prostate cancer. In addition, we found that CNAs of *FOXP2* were prominently enriched in ETS fusion-negative prostate tumors from the MSKCC/DFCI dataset (n = 685) (*Armenia et al., 2018*) (p=2.42 × 10$^{-6}$, by Fisher's exact test) (*Figure 1—figure supplement 4F*), suggesting that *FOXP2* CNAs and ETS fusions were partially mutually exclusive. Similar to our observation, Stumm et al. reported moderate to strong FOXP2 protein expression in 75% of prostate tumors, and a higher protein expression level of FOXP2 was correlated with higher Gleason score, advanced T stage, and earlier cancer recurrence in *ERG* fusion-negative prostate cancers (*Stumm et al., 2013*).

Here, we identified a novel recurrent *FOXP2-CPED1* fusion in two localized prostate tumors. Due to loss of miR-27a/b-mediated transcriptional regulation, the fusion yielded a truncated FOXP2 protein

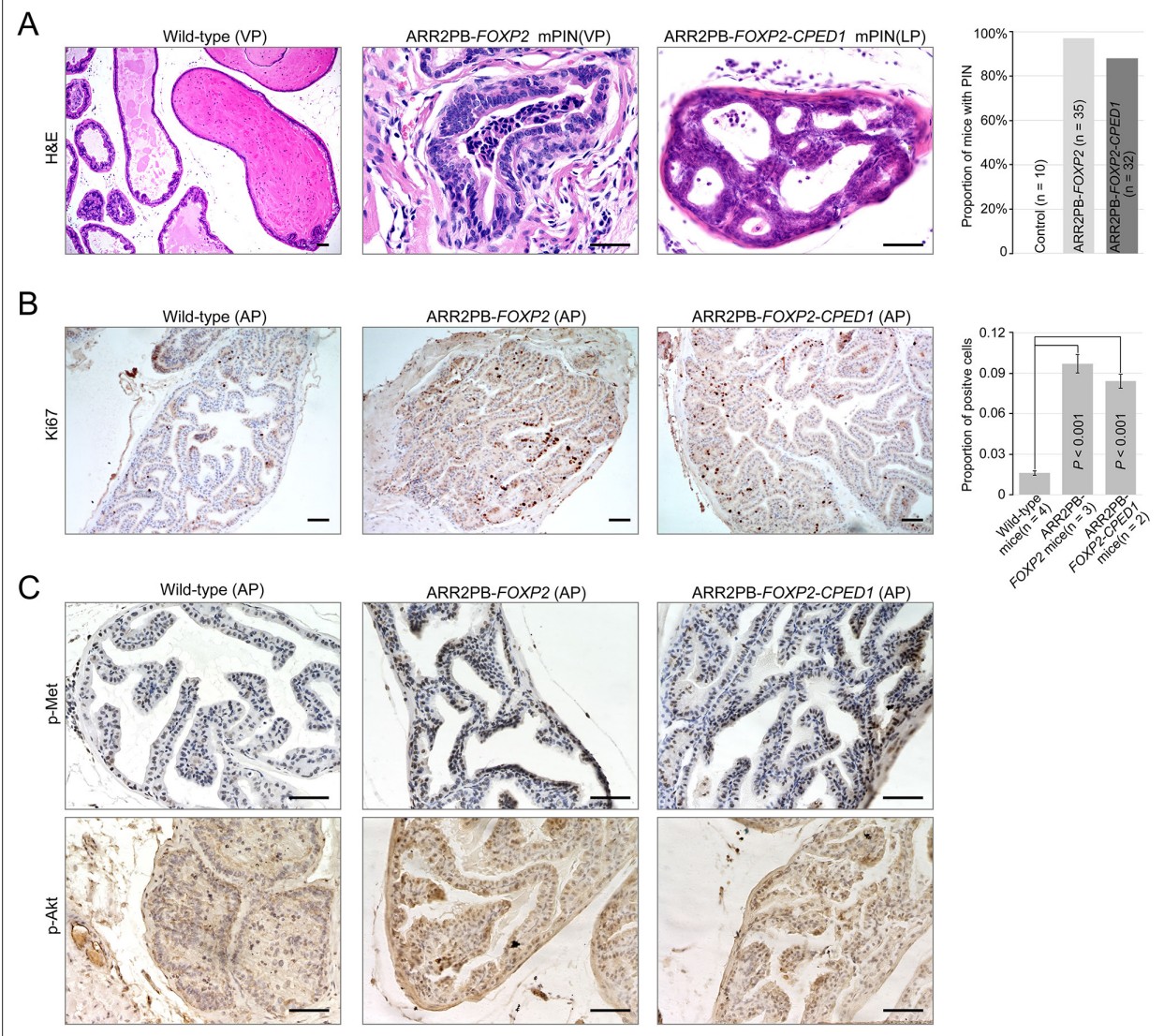

**Figure 4.** Prostate-specific overexpression of *FOXP2* causes prostatic intraepithelial neoplasia. (**A**) Histological images of prostatic intraepithelial neoplasia in ARR2PB-*FOXP2* or ARR2PB-*FOXP2-CPED1* mice (×200) compared to wild-type control (×40) (mice 14 mo of age). A bar graph showing the incidence of mPIN in transgenic mice at an average of 14 mo of age. See **Supplementary file 1e** for details. (**B**) Immunohistochemistry of Ki67 in prostate glands reveals a significant increase in proliferation for ARR2PB-*FOXP2* or ARR2PB-*FOXP2-CPED1* mice at 6 mo of age (×100). The bar graph shows the proportion of Ki67-positive cells per gland (mean ± SD) reported for at least three representative prostate glands per mouse. p-Values were calculated by two-tailed Student's *t*-test. (**C**) Immunohistochemical analysis of Met (Y1234/1235) (upper) and Akt (S473) (lower) activity in prostate glands shows the upregulation of Met signaling in ARR2PB-*FOXP2* and ARR2PB-*FOXP2-CPED1* mice (×200).

The online version of this article includes the following source data and figure supplement(s) for figure 4:

**Figure supplement 1.** Generation of transgenic mice with prostate-specific expression of *FOXP2* or *FOPX2-CPED1*.

**Figure supplement 1—source data 1.** Uncropped gel for *Figure 4—figure supplement 1C*.

**Figure supplement 1—source data 2.** Uncropped blot for *Figure 4—figure supplement 1D*.

that was highly expressed in the fusion-carrying tumor (*Figure 1—figure supplement 1F–L*). In addition, whole-genome sequencing of the fusion-positive tumor suggested that the *FOXP2* fusion was an early event in the tumor (*Figure 1—figure supplement 1D*). The truncated FOXP2 protein retained the full forkhead DNA binding domain (DBD). In humans, mutations in the DBD of FOXP2 cause severe autosomal-dominant disorders of speech and language (*Lai et al., 2001*). Several lines of evidence also showed that the DBD of some other FOX family proteins was crucial for the growth and invasion of tumors (*Gormally et al., 2014*; *Huang et al., 2015*). Consistent with the observation of cellular

transformation resulting from *FOXP2* overexpression, ectopically expressed *FOXP2-CPED1* induced the transformation of normal or benign prostate cells to malignant cells in vitro and in vivo and aberrantly activated oncogenic MET signaling, suggesting that the oncogenic role of the FOXP2 protein could be DBD dependent. Taken together, these findings for the *FOXP2* fusion further supported that *FOXP2* has an oncogenic potential.

To identify genes that are significantly regulated by *FOXP2*, we compared two sources of differential gene expression data from TCGA primary prostate tumors and our *FOXP2*-transformed NIH3T3 cells (*Figure 2—figure supplement 1E*). We found 60 common genes whose expression was significantly correlated with *FOXP2* expression. The overlapping genes included five known cancer driver genes (including *HGF*, *MET,* and *PIK3R1*) (*Figure 2D*, *Figure 2—figure supplement 1C and E*) and five putative *FOXP2* targets (*Figure 2—figure supplement 1E*), such as *MET* and *PIK3R1*, identified by chromatin immunoprecipitation assays in both mice and humans (*Spiteri et al., 2007*; *Xu et al., 2022*; *Mukamel et al., 2011*; *Vernes et al., 2011*; *Vernes et al., 2007*). Our analysis of the CUT&Tag data supported that *MET* is a target of FOXP2.

Immunohistochemical staining of normal and malignant human prostate samples showed that MET protein expression is absent in epithelial cells of normal prostate glands and low in benign prostate hyperplasia, whereas it is frequently detected in PIN, localized and metastatic prostate tumors (*Nakashiro et al., 2003*; *Pisters et al., 1995*). Transgenic overexpression of Met in mouse prostate epithelial cells is sufficient to induce PIN under HGF conditions and markedly promotes prostate tumor initiation, invasion and metastasis in a Pten-deficient background (*Mi et al., 2018*). Our data showed that overexpression of *FOXP2* activated oncogenic MET signaling in *FOXP2*-transformed prostate epithelial cells and human prostate tumors, and inhibition of MET signaing activation in *FOXP2*-overexpressing RWPE-1 cells and NIH3T3 cells significantly suppressed the anchorage-independent growth or foci formation of these cells. In agreement with our findings, treatment of both PC3 and LNCaP prostate tumor cells with a MET inhibitor resulted in reduced cell proliferation (*Tu et al., 2010*).

MET is a receptor tyrosine kinase that is activated by its ligand, hepatocyte growth factor (HGF), and this activation triggers transphosphorylation of MET (*Bardelli et al., 1999*; *Bottaro et al., 1991*; *Cooper, 1992*; *Hov et al., 2004*). In our study, we found a positive correlation between the expression of both *MET* and *HGF* and the expression of *FOXP2* in prostate cancer tissues, *FOXP2*-overexpressing human prostate epithelial cells RWPE-1, and *FOXP2*-overexpressing NIH3T3 cells. Additionally, we observed that overexpression of *FOXP2* resulted in an increased phosphorylation level of MET in RWPE-1 cells, while knockdown of *FOXP2* resulted in a decreased phosphorylation level of MET in PC3 and LNCaP cells. Furthermore, we identified potential FOXP2-binding fragments in both *MET* and *HGF* using the CUT&Tag method. Based on these findings, we speculate that the increased phosphorylation of MET may be due to the increased expression of MET itself and the increased expression of its ligand HGF.

Phase III clinical trials have shown that the MET kinase inhibitor cabozantinib has some clinical activity in patients with advanced prostate cancer, including conferring improvements in the bone scan response, radiographic progression-free survival, and circulating tumor cell conversion, although it failed to significantly improve overall survival (*Smith et al., 2016*; *Sonpavde et al., 2020*). *Qiao et al., 2016* provided experimental evidence that MET inhibition was only effective in prostate cancer cells with MET activation. Our data showed that overexpressed *FOXP2* aberrantly activated oncogenic MET signaing in transformed prostate epithelial cells and human prostate tumors. We observed that *FOXP2*-overexpressing cells were more sensitive to inhibitors of the MET signaing pathway than control cells. Collectively, these data suggested a potential therapeutic option for prostate cancer patients with high expression levels of *FOXP2*. Future work is required to determine whether additional pathways are activated or inhibited by *FOXP2*. In summary, our data indicated that *FOXP2* possesses an oncogenic potential and is involved in tumorigenicity of prostate. Aberrant *FOXP2* expression activates MET signaing pathway in prostate cancer with potential therapeutic implication.

## Materials and methods
### Clinical specimen
One hundred localized primary prostatic adenocarcinomas that underwent radical prostatectomy were collected. All tumor specimens were independently evaluated by two pathologists for histological

diagnosis and Gleason score on hematoxylin and eosin (H&E) stained slides. The presence of or morphological absence of adenocarcinoma was determined by one pathologist through review of the 4 μm H&E stained tissue slice, and then high-density cancer foci (more than 70% tumor cellularity) and contamination-free normal prostate tissues were cut into ~1 mm³ tissue blocks for extraction of DNA, RNA, or protein. Ten pairs of prostate tumors (PC_1 to PC_10) and matched normal tissues (NT_1 to NT_10) were used for transcriptome sequencing (*Supplementary file 1a*). Additionally, whole-genome sequencing was performed on the *FOXP2-CPED1* fusion-positive tumor (PC_1). Reduced Representation Bisulfite Sequencing (RRBS) analysis and small RNA sequencing were performed on PC_1 and NT_1.

## Cell lines and cell culture

NIH3T3 mouse primary fibroblast cells, the immortalized human prostate epithelial cell line RWPE-1, prostate cancer cell line LNCaP and PC3 were obtained from the Cell Resource Center, Peking Union Medical College, National Infrastructure of Cell Line Resource. Prostate cancer cell line VCaP and human embryonic kidney (HEK) 293T cells were obtained from ATCC. The identity of the cell lines was authenticated with STR profiling. The cell lines were checked free of mycoplasma contamination by PCR. LNCaP cells and HEK 293T cells were cultured in RPMI-1640 (Life Technologies) supplemented with 10% FBS (Life Technologies), L-glutamine (2 mM) (Life Technologies), penicillin (100 U/ml) (Life Technologies), and streptomycin (100 μg/ml) (Life Technologies). PC3 cells were cultured in F-12K (Life Technologies) supplemented with 10% FBS (Life Technologies), L-glutamine (2 mM) (Life Technologies), penicillin (100 U/ml) (Life Technologies), and streptomycin (100 μg /ml) (Life Technologies). NIH3T3 cells and VCaP cells were maintained in DMEM (Life Technologies) supplemented with 10% FBS, L-glutamine (2 mM), penicillin (100 U/ml), and streptomycin (100 μg /ml). RWPE-1 cells were grown in Defined Keratinocyte-serum free medium (Defined Keratinocyte-SFM, Life Technologies) with L-glutamine (2 mM), penicillin (100 U/ml), and streptomycin (100 μg /ml). All cells were grown at 37°C.

## RNA sequencing of localized prostate cancer samples

Total RNAs of 10 pairs of tumors (PC_1 to PC_10) and matched normal tissues (NT_1 to NT_10) were extracted with Trizol (Life Technologies). RNA integrity number (RIN) > 7.0 and a 28S:18S ratio >1.8. Sequencing libraries for strand-specific transcriptome was carried out as described previously (*Park-homchuk et al., 2009*) by BGI-Shenzhen (Shenzhen, China). Briefly, Beads containing oligo (dT) were used to isolate poly (A) mRNA from total RNA. The mRNA was fragmented into short fragments by the fragmentation buffer. Using these short fragments as templates, random hexamer-primers were used for synthesization of the first-strand cDNA. After purification with the G-50 gel filtration spin-column (GE Healthcare Life Sciences) to remove dNTPs, second-strand synthesis was performed by incubation with RNase H, DNA polymerase, and dNTPs containing dUTP (Promega). A single 3' 'A' base was added to the end-repaired cDNA. Upon ligation with the Illumina PE adaptors, the products were gel-recovered and subsequently digested with N-glycosylase (UNG; Applied Biosystems) to remove the second-strand cDNA. Samples were then amplified with Phusion polymerase and PCR primers of barcode sequence. The amplified library was sequenced on an Illumina HiSeq 2000 sequencing platform. The paired-end reads obtained from HiSeq 2000 were aligned to the human reference genome and transcriptome (hg19) using SOAP2 program (*Li et al., 2009b*). No more than five mismatches were allowed in the alignment for each read. The gene expression level was calculated by using RPKM (*Mortazavi et al., 2008*) (reads per kilobase transcriptome per million mapped reads), and the formula is shown as follows:

$$RPKM = \frac{10^6 C}{NL/10^3}$$

Given to be the expression of gene A, $C$ represents number of reads that are uniquely aligned to gene A, $N$ represents total number of reads that are uniquely aligned to all genes, and $L$ represents number of bases on gene A. Using 'The significance of digital gene expression profiles' (*Audic and Claverie, 1997*), we identified differentially expressed genes between the tumors and matched normal samples based on the following criteria: FDR ≤ 0.001 and fold change ≥ 1.5.

### Detecting human fusion genes in localized prostate cancer samples

We used SOAPfuse (*Jia et al., 2013*) to detect gene fusion events from RNA-seq data based on the default parameters. After mapping RNA-seq reads to the human reference genome sequence (hg19) and Ensembl annotated genes (release 64), SOAPfuse sought span-reads and junc-reads to support fusion detection: paired-end reads that mapped to two different genes were defined as span-reads, and reads covering the junction sites were called as junc-reads. SOAPfuse detects fusion events generated by genome rearrangements with breakpoints in intron and exon regions.

### Small RNA sequencing of localized prostate cancer sample

Total RNAs of the tumor (PC_1) and matched normal tissue (NT_1) were extracted with Trizol (Life Technologies). Preparation of small RNA library was performed according to the manufacturer's instructions (Preparing Samples for Analysis of Small RNA, Part #11251913, Rev. A) by BGI-Shenzhen (Shenzhen, China). Briefly, small RNA sequence ranging from 18 to 30 nt was gel-purified and ligated to the Illumina 3′ adaptor and 5′ adaptor. Ligation products were then gel-purified, reverse transcribed, and amplified. The amplified library was sequenced on an Illumina HiSeq 2000 platform. The small RNA reads were subjected to the following filtering processes: (i) filtering out low-quality reads; (ii) trimming 3′ adaptor sequence; (iii) removing adaptor contaminations resulted from adaptor ligation; and (iv) retaining only short trimmed reads of sizes from 18 to 30 nt. To annotate and categorize small RNAs into different classes, we filtered out small RNA reads that might be from known noncoding RNAs by comparing them with known noncoding RNAs, including rRNA, tRNA, snRNA, and snoRNA, which were deposited in the Rfam database and the NCBI GenBank. Small RNA reads assigned to exonic regions were also discarded. After removing small RNA reads in term of the above categories, the rest were subjected to MIREAP, which identified miRNA candidates according to the canonical hairpin structure and sequencing data. The identified small RNA (miRNA) reads were then aligned to miRNA reference sequences with tolerance for one mismatch. Reads that were uniquely aligned and overlapped with known miRNAs were considered as candidate miRNAs.

### Whole-genome sequencing of primary prostate cancer sample

Genomic DNA of PC_1 was extracted with phenol-chloroform method and subjected to whole-genome sequencing by BGI-Shenzhen (Shenzhen, China). After removing reads that contained sequencing adapters and low-quality reads with more than five ambiguous bases, high-quality reads were aligned to the NCBI human reference genome (hg19) using BWA (v0.5.9) (*Li and Durbin, 2009*) with default parameters. Picard (v1.54) was used to mark duplicates and followed by Genome Analysis Tool kit (v1.0.6076, GATK IndelRealigner) (*McKenna et al., 2010*) was followed to improve alignment accuracy. The SNVs were detected by SOAP snp (*Li et al., 2009a*) and several filtering steps were performed to reduce the false positives, including the removal of SNVs whose consensus quality was lower than 20, single-nucleotide variations (SNVs) located within 5 bp of the splice donor sites, and SNVs with less than three spanning reads. We next used BreakDancer to detect structural variations (SVs). After SVs were identified, we used ANNOVAR to do annotation and classification.

### Reduced representation bisulfite sequencing of primary prostate cancer sample

Genomic DNAs of PC_1 and NT_1 were subjected to RRBS by BGI-Shenzhen (Shenzhen, China). Before library construction, MspI (NEB R0106L) was used to digest the genomic DNA. Next, the Illumina Paired-End protocol was used to construct the library. DNA libraries of 40–220 bp were excised. Then the excised DNA was recovered by columns, purified by MiniElute PCR Purification Kit (QIAGEN, #28006), and eluted in EB. Bisulfite was converted using the EZ DNA Methylation-Gold kit (ZYMO). All the bisulfite-converted products were amplified by PCR in a final reaction. PCR products were purified and recovered, followed by sequencing with an Illumina HiSeq 2000 platform. The RRBS reads were aligned to human genome reference (hg19) using SOAP2 allowing less than two mismatches. We then used these uniquely mapped reads that contained the enzyme cutting site to get methylation information of cytosine as described previously (*Wang et al., 2012*). Methylation level was determined by dividing the number of reads covering each methyl-cytosine by the total reads that covered cytosine, and the methylation information for each gene were calculated when the promoters of which were covered by at least 5 CpGs.

## RT-PCR, qPCR, 3′RACE-PCR, and long-range PCR

Total RNAs were extracted from the human prostate tumors and the normal tissues and various cell lines using Trizol (Life Technologies). Reverse transcription was performed using the M-MLV reverse system (Takara) to obtain cDNA. qPCR was performed using with the SYBR Premix Ex Taq mix (TaKaRa) according to the manufacturer's instructions, and the samples were run on an iQ5 Multicolor Real-time PCR Detection System (Bio-Rad). Results were normalized to expression levels for reference genes (*GAPDH* or *EIF4H*). The relative expression levels of genes were calculated using the ΔΔCT method. 3′RACE kit (Takara) was used to convert RNAs of the prostate tumors into cDNAs by a reverse transcriptase and oligo-dT adapter primer. The cDNAs were amplified by using *FOXP2*-specific primers, which annealed to exon 6, exon 11, and exon 16, and an oligo-dT adapter primer, respectively. Amplified fragments of the putative sizes were subjected to Sanger sequencing. Genomic DNAs of the fusion-positive tumors (PC_1 and PC_11) were used for long-range PCR. Amplified fragments of the putative sizes were subjected to Sanger sequencing. The primers for RT-PCR, qPCR, 3′RACE-PCR, and long-range PCR are available in *Supplementary file 1f*.

## Plasmid construction

Constructs of *FOXP2* (Origene, #RC215021, NM_014491) and *CPED1* (Origene, #RC217158, NM_024913) were purchased. The CDS of *FOXP2-CPED1* fusion, truncated *FOXP2* were amplified from the corresponding fusion-positive tumor and then cloned into pFlag-CMV4 vector (Sigma, #E7158). To construct the *FOXP2-CPED1* fusion containing the 3′UTR of *FOXP2*, the 3′UTR of *FOXP2* was generated from the *FOXP2* 3′UTR plasmid (Origene, #SC212500, NM_014491) by PCR. The PCR product was fused into *FOXP2-CPED1* cDNA by overlapping PCR and then the fragment of *FOXP2-CPED1*+ 3′UTR was cloned into pFlag-CMV4 vector. To construct the plasmids for luciferase assay, the 3′UTR of *FOXP2* was amplified from *FOXP2* 3′UTR construct and ligated into a pmirGlo Dual-luciferase miRNA Target Expression Vector (Promega) to form 3′UTR-luciferase reporter vector. The full predicted seed sequences of miR-27a and miR-27b on 3′UTR of *FOXP2* were deleted from the 3′UTR-luciferase reporter vector using overlapping PCR method to create a mutant *FOXP2* 3′UTR construct. The CDS of *FOXP2-CPED1* or *FOXP2* was cloned into the pCDH-CMV-MCS-EF1 Lentivector (SBI, #CD513B-1). The CDS of *MET* (NM_000245) was cloned into the PCDNA3.1.HA vector. To knockdown of endogenous *FOXP2* expression in PC3 and LNCaP cells, four shRNA fragments (1# ggaagacaatggcattaaacattcaagagatgtttaatgccattgtcttcctttttt; 2# ggacagtcttcagttctaagtttcaagagaacttagaactgaagactgtcctttttt; 3# gcaggtggtgcaacagttagattcaagagatctaactgttgcaccacctgcttttt; 4# gcgaacgtcttcaagcaatgattcaagagatcattgcttgaagacgttcgctttttt) targeting the exon 7, exon 9, and exon 10 of *FOXP2* coding sequencing were cloned into pLent-4in1shRNA-GFP vector. The Scrambled-pLent-4in1shRNA-GFP control vector included a fully scrambled sequence. To knockdown of endogenous *MET* expression in RWPE-1 cells, si*MET* (5′-caatcatactgctgacata-3′) was used. The primers used for various plasmid constructions are listed in *Supplementary file 1f*.

## Lentivirus transduction and establishment of stable cell lines

Lentiviruses were produced by cotransfecting the pCDH-*FOXP2* construct or pCDH-*FOXP2-CPED1* construct or pCDH vector or *FOXP2*-pLent-4in1shRNA-GFP vector or Scrambled-pLent-shRNA-GFP vector with the packaging plasmid Mix pPACK (SBI) in HEK293T cells. At 36 hr post-transfection, viral supernatants were collected, centrifuged at 12,000 × *g* for 20 min, filtered through 0.45 µm Steriflip filter unit (Millipore), and concentrated using ViraTrap lentivirus purification kit (Biomiga). NIH3T3 or RWPE-1 cells or PC3 or LNCaP at 90% confluence were infected with Lenti-*FOXP2*, Lenti-*FOXP2-CPED1*, Lenti-pCDH, Lenti-*FOXP2*-pLent-4in1shRNA-GFP and Lenti-scrambled-pLent-shRNA-GFP viruses, respectively. Cells were spun at 1800 × *g* for up to 45 min at room temperature. After 24 hr incubation at 37°C, cells were split and placed into selective medium containing 1 µg/ml puromycin. Puromycin-resistant clones were grown from single cell. Western blot using anti-FOXP2 antibody (Millipore, #MABE415, 1:1000 dilution) was performed to detect FOXP2 protein expression in *FOXP2* or *FOXP-CPED1*-overexpressing NIH3T3 or RWPE-1 cell clones or sh*FOXP2*-expressing PC3 or LNCaP cell clones.

## Focus formation assay

NIH3T3 cells stably expressing the *FOXP2-CPED1*, *FOXP2,* and lentivirus pCDH vector, respectively, and parental cells were seeded at a concentration of $1 \times 10^4$ cells (parental NIH3T3 cells were mixed with NIH3T3 cells overexpressing *FOXP2-CPED1*, *FOXP2,* or lentiviral vector, respectively, at a 100:1 cell ratio) per well in a 6-well plate. Cells were then cultured for 14–16 d. The representative foci were either taken pictures using a Nikon microscope or fixed with 4% paraformaldehyde solution for 10 min and subsequently stained with 0.05% crystal violet and solubilized with 4% acetic acid. RWPE-1 cells stably expressing the *FOXP2-CPED1*, *FOXP2,* and lentiviral vector, respectively, and parental cells were seeded at a concentration of $1 \times 10^4$ cells per well in a 6-well plate. Cells were then cultured for 14–16 d, then fixed with 4% paraformaldehyde solution for 10 min and subsequently stained with 0.05% crystal violet. The representative foci were taken pictures using a Nikon microscope. PC3 cells stably expressing shRNA targeting *FOXP2* or parental cells were plated at a density of 1000 cells per well in triplicate in a 6-well plate for 10 d. LNCaP stably expressing shRNA targeting *FOXP2* cells or parental cells were plated at a density of 1000 cells per well in triplicate in a 6-well plate for 25 d, then fixed with 4% paraformaldehyde solution for 10 min and subsequently stained with 0.05% crystal violet.

## Soft agar colony formation assay

For the assay of NIH3T3 cells stably expressing the *FOXP2-CPED1*, *FOXP2,* lentiviral vector, or parental cells, cells were collected and suspended in 0.4% soft agar at 1000 cells (cell suspension in complete growth medium mixed with 2× DMEM, 20% FBS, and 1.2% agar [Difco Noble Agar, BD Biosciences, #214230] at a 1:1:1 dilution). The cell suspension (0.75 ml) was added to each well of a 24-well plate and kept at 4°C for 20 min. For RWPE-1 cells stably expressing the *FOXP2-CPED1*, *FOXP2,* lentiviral vector, or parental RWPE-1, cells were suspended in 0.4% soft agar at 2000 cells (cell suspension in complete growth medium mixed with Defined Keratinocyte-SFM with 10% FBS and 1.2% agar at a 2:1 dilution). The cell suspension (0.75 ml) was added to each well of a 24-well plate and kept at 4°C for 20 min. Each cell line was plated in quadruplicate. The fresh mediums were changed every 4 d and incubated for 4 wk. The colony counting was performed using a Nikon microscope and representative images were then acquired.

## Cell proliferation assay, IC$_{50}$ assay, and drug response assay

For growth curve of NIH3T3 and RWPE-1 cell lines, indicated cells were seeded at 3000 cells per well in a 96-well plate in quadruplicate, treating with increasing drug concentrations for 48 hr. Foretinib (#A2974) was purchased from ApexBio. MK2206 (#HY-10358) was purchased from MCE. The viability rate of each cell line treated with different drug concentrations was normalized to that of the corresponding untreated cells. The in vitro half-maximal inhibitory concentration (IC$_{50}$) values were determined using Orange 8.0 software and dose–response curves were plotted with package drc in R environment. For proliferation assay of RWPE-1, PC3, or LNCaP cells, indicated cells were seeded at 3000 cells per well in a 96-well plate in quadruplicate for 6 d (RWPE-1), 8 d (PC3), and 9 d (LNCaP), respectively. The viability of cells was determined using a Celltiter-Glo assay (Promega) on INFINITE 200 Pro multimode reader (TECAN). For drug response assay, NIH3T3 cells stably expressing the *FOXP2* and *FOXP2-CPED1*, respectively were suspended in 0.4% soft agar at 3000 cells (cell suspension in complete growth medium mixed with 2× DMEM, 20% FBS, and 1.2% agar at a 1:1:1 dilution) with indicated drug concentrations. Each cell line was plated in quadruplicate. The medium with indicated drug concentration was changed every 4 d and incubated for 4 wk. The colony counting was performed using a Nikon microscope and representative images were then acquired. RWPE-1 cells stably expressing the *FOXP2* and *FOXP2-CPED1*, respectively, were plated at a density of 3000 cells per well in triplicate in a 6-well plate for 12 d with indicated drug concentrations, then fixed with 4% paraformaldehyde solution for 10 min and subsequently stained with 0.05% crystal violet.

## Animal xenograft studies

Fifty 6–8-week-old non-obese diabetic severe combined immune deficiency spontaneous female mice (NOD.CB17-*Prkdc*scid/NcrCrl) were used. $2 \times 10^6$ NIH3T3 cells in PBS were injected subcutaneously into the flanks of NOD-SCID mice (≥5 mice for each cell line) at an inoculation volume of 100 µl with

a 23-gauge needle. Mice were monitored for tumor growth, and 2 mo was selected as an endpoint. Tumor volumes (*V*) were calculated with the following formula: $((\text{width})^2 \times (\text{length}))/2 = V$ (cm$^3$).

## Protein blot analysis

Various cell lines, the frozen human tumors and normal tissues, and the tumors from animal xenograft studies were lysed in RIPA lysis buffer containing protease and phosphatase inhibitors (Roche Inc), and sonicated for 20 s. These extractions were resolved by SDS-PAGE and electrotransferred to PVDF membranes (Millipore). Membranes were blocked for 1 hr and blotted for various primary antibodies overnight in 5% non-fat milk or 5% BSA in Tris buffered saline solution, 0.5% Tween-20 (TBST) (Thermol Scientific). The following primary antibodies were used: anti-FOXP2 (Millipore, #MABE415, 1:1000, an antibody to N terminus of FOXP2), anti-FLAG tag (M2) (Sigma, #F1804, 1:1000), anti-Phospho-Akt (Ser473) (Cell Signaling Technology, #9271, 1:1000), anti-Akt (Cell Signaling Technology, #9272, 1:1000), anti-Phospho-Met (Tyr1234/1235) (Cell Signaling Technology, #3077, 1:1000), anti-Met (Abcam, #ab51067, 1:1000), anti-Androgen receptor (Abcam, #ab133273, 1:1000), anti-beta-Actin (C4) (Santa Cruz Technology, #sc-47778, 1:1000), and anti-alpha Tubulin (DM1A) (Abcam, #ab7291, 1:5000). Horseradish-preoxidase conjugated antibodies to mouse (Abcam, #ab6789, 1:5000) or rabbit (Abcam, #ab6721, 1:5000) were used as the secondary antibodies, and Immobilon Western Chemiluminescence HRP Substrate (Millipore) was used for detection.

## Transfection of microRNAs and luciferase reporter assay

The microRNA mimics and microRNA inhibitors synthesized by Ribobio were used as follows: hsa-miR-19b-5p, hsa-miR-23a-3p, hsa-miR-23b-3p, hsa-miR-27a-3p, has-miR-27b-3p, hsa-miR-132-3p, hsa-miR-134-5p, hsa-miR-186-5p, hsa-miR-196b-5p, hsa-miR-212-5p, hsa-miR-214-3p, hsa-miR-379-5p, microRNA negative control (NC), hsa-miR-27a-3p inhibitor, hsa-miR-27b-3p inhibitor, and negative control inhibitors (NC-I). The HEK293T cells were seeded in 96-well plate in quadruplicate and co-transfected with each of 12 microRNAs and the wild-type *FOXP2* 3′UTR firefly luciferase reporter construct using Lipofectamine 2000 (Life Technologies) for 26 hr. Wild-type or mutant *FOXP2* 3′UTR firefly luciferase reporter construct was co-transfected into HEK293T cells with NC or miR-27a or miR-27b. After 26 hr, cell lysates were prepared to consecutively measure the firefly and *Renilla* luciferase activities using Dual-luciferase reporter assay system (Promega) on INFINITE 200 Pro multimode reader (TECAN). The firefly luciferase activity was normalized by *Renilla* luciferase activity.

## Histological analysis

Mice xenograft tumors originating from NIH3T3 cell lines that expressed *FOXP2* or *FOXP2-CPED1* were fixed in 4% paraformaldehyde. Consecutive paraffin sections of the mice xenograft tumors and prostates of ARR2PB-*FOXP2* or ARR2PB-*FOXP2-CPED1* transgenic mice (4 μm thickness) were used for H&E staining and immunohistochemical analyses. Specimens from 25 cases of benign prostatic hyperplasia and 45 cases of primary prostate cancer were analyzed for FOXP2 staining with an immunohistochemical assay. Protein expression was evaluated as negative, low, medium, and strong. The sections were pretreated with citrate buffer (pH 9.0) in a microwave oven for 20 min for antigen retrieval. The primary antibody to FOXP2 (Sigma, #HPA000382) was used at a 1:200 dilution. The following primary antibodies were used for immunohistochemical staining: anti-Ki67 (Abcam, #ab16667, 1:300), anti-Phospho-Akt (Ser473) (Cell Signaing Technology, #9271, 1:200), and anti-Phospho-Met (Tyr1234/1235) (Cell Signaling Technology, #3077, 1:200).

## Pathway enrichment analysis

Gene set enrichment analysis of the 3206 *FOXP2* expression-correlated genes in human prostate tumors was performed by GSEA (v2.0) (*Subramanian et al., 2005*) using gene sets from Molecular Signatures Database (MSigDB) (v6.2) (http://software.broadinstitute.org/gsea/msigdb/index.jsp). Cancer-associated pathway analysis of the 3206 *FOXP2* expression-correlated genes was performed using gene sets from MSigDB and KEGG pathway database, and results were visualized using R (v3.3.1).

## Cleavage under targets and tagmentation (CUT&Tag)

A total of 100,000 LNCaP cells with overexpression of HA-tagged *FOXP2* were used. The CUT&Tag assay and subsequent DNA library construction were performed by following the manufacturer's instructions for the Hyperactive Universal CUT&Tag Assay Kit for Illumina (Vazyme, #TD903-01). An anti-HA Tag polyclonal antibody (Invitrogen, #PA1-985) and control rabbit IgG (ABclonal, #AC005) were used as primary antibodies. Libraries were tested by an Agilent 2100 bioanalyzer. Paired-end sequencing was performed on an Illumina Novaseq 6000 system. Paired-end reads (150 bp) were aligned using Bowtie2 version 2.2.5. Macs2 peak calling software was used for peak calling. Visualization of CUT&Tag peaks was performed in Integrative Genomics Viewer. The binding sites of *HGF* and *MET* were validated by PCR and Sanger sequencing using the CUT&Tag libraries as the DNA template.

## Establishment of ARR2PB-*FOXP2* and ARR2PB-*FOXP2-CPED1* transgenic mice

To generate a mouse model expressing *FOXP2* or *FOXP2-CPED1* in a prostate-specific manner, we amplified the human *FOXP2* or *FOXP2-CPED1* cDNA sequence from the PC_1 tumor harboring the *FOXP2-CPED1* fusion by RT-PCR. A composite ARR2PB promoter (*Zhang et al., 2000*) and a hGH_PA_terminator were introduced into the two sequence to produce ARR2PB-*FOXP2*-hGH_PA and ARR2PB-*FOXP2-CPED1*-hGH_PA by overlapping PCR method. Subsequently, they were cloned into a pBluescript SK+(PBS) expression vector that contains a T7 promoter to obtain PBS-ARR2PB-*FOXP2* and PBS-ARR2PB-*FOXP2-CPED1* using KpnI and XhoI multicloning sites. The ARR2PB-*FOXP2* and ARR2PB-*FOXP2-CPED1* mRNA were generated in vitro using the MEGA shortscript T7 kit and were delivered into mouse zygotes by microinjection, respectively, to establish transgenic mice (C57/BL6 background). Next, mice were analyzed for construct integration by PCR genotyping assay (*Supplementary file 1f*). In total, we identified five founder positive lines for the *FOXP2* transgene (three positive) and the *FOXP2-CPED1* transgene (two positive). Subsequently, we analyzed the future offspring using PCR genotyping assay. We evaluated the expression of human *FOXP2* and *FOXP2-CPED1* at the mRNA and protein levels by RT-PCR, western blot, and immunohistochemical staining. All the five founder lines showed mRNA and protein expression of human *FOXP2* and *FOXP2-CPED1* in all lobes of the transgenic mice prostate. Founder lines 16# and 48# for the *FOXP2* transgene and founding line 26# for the *FOXP2-CPED1* transgene were subsequently used for phenotypic analysis.

## Statistics

We used SPSS (v16.0), Origin (v8.0), or R (v3.3.1) (The R Project for Statistical Computing, http://www.r-project.org/) software for statistical calculation. Specific statistical tests, number of samples, and experimental or public data utilized in each analysis are shown along the main text or in the figure legends. Cell culture-based experiments were conducted three times or more, which were triplicated or quadruplicated. The data are presented as mean ± SD. Student's *t*-test was used to compare the difference between means in normal distributions. In the boxplots, boxes display the 25th to 75th percentiles, lines represent the medians, and whiskers represent 1.5× the interquartile range. Mann–Whitney *U*-test was used to compare the difference between data in non-normal distributions. The two-tailed $p < 0.05$ was considered to be statistically significant. Spearman rank correlation was used to measure the association between expressions of individual genes. We used the Cochran–Mantel–Haenszel test to estimate adjusted odds ratio ($OR_{MH}$) and adjusted p-values when *ETS*-fusions were considered a confounding factor.

## Data and software availability

The RNA-seq data from 10 pairs of primary prostate tumors and normal tissues has been deposited in the GEO database with the accessions codes: GSE114740. The whole-genome sequencing data from *FOXP2-CPED1* fusion-positive tumor (PC_1) has been deposited in the SRA database with the accessions: SRR7223723. All other available public data supporting the findings of this study can be found in the article or its supplementary files obtained from the cBioPortal database (http://www.cbioportal.org/), the GTExPortal database (https://gtexportal.org/home/documentationPage), and the Gene Expression Omnibus database (https://www.ncbi.nlm.nih.gov/geoprofiles). We considered the prostate cancer samples with *FOXP2* mRNA expression level <0.5 normalized RNA-seq by

expectation maximization (RSEM) from the TCGA dataset (Prostate Adenocarcinoma, Provisional) as *FOXP2*-negative tumors. URLs.: Rfam database, http://rfam.xfam.org/; NCBI Genbank, http://www.ncbi.nlm.nih.gov/genbank/; IREAP, https://sourceforge.net/projects/mireap/; miRNA reference sequences, http://www.mirbase.org/, release 19; Break Dancer, http://breakdancer.sourceforge.net/; TargetScan, http://www.targetscan.org/vert_71/; microRNA.org, http://www.microrna.org/; PICTAR5, http://pictar.mdc-berlin.de/; Cancer Gene Census, https://cancer.sanger.ac.uk/census.

## Acknowledgements

We thank Mingming Shi, Qiang Gao, and Xuanmin Guang for helpful advice and assistance with analyses. The work was supported by the National Natural Science Foundation of China grants (81872096) (to X Zhu), (81541152) (to Y Zhao), 81472408 (to JW), 81570789 (to JL), the Chinese Academy of Medical Sciences Innovation Fund for Medical Sciences (2021-I2M-1-050) (to Y Zhao); the CAMS Innovation Fund for Medical Sciences (CIFMS) (2018-I2M-1-002) (to Y Zhao), 12th 5-year National Program form the Ministry of Scientific Technology 2012BAI10B01 (to JW and ZY), 973 program grants from the National Basic Research Program of China 2014CB910503 (to JL).

## Additional information

### Funding

| Funder | Grant reference number | Author |
|---|---|---|
| National Natural Science Foundation of China | 81872096 | Xiaoquan Zhu |
| National Natural Science Foundation of China | 81541152 | Yanyang Zhao |
| National Natural Science Foundation of China | 81472408 | Jianye Wang |
| National Natural Science Foundation of China | 81570789 | Jian Li |
| Chinese Academy of Medical Sciences Initiative for Innovative Medicine | 2021-I2M-1-050 | Yanyang Zhao |
| Chinese Academy of Medical Sciences Initiative for Innovative Medicine | 2018-I2M-1-002 | Yanyang Zhao |
| Ministry of Scientific Technology | 12th 5-year National Program - 2012BAI10B01 | Ze Yang<br>Jianye Wang |
| National Basic Research Program of China | 973 Program Grants - 2014CB910503 | Jian Li |

The funders had no role in study design, data collection and interpretation, or the decision to submit the work for publication.

### Author contributions

Xiaoquan Zhu, Conceptualization, Data curation, Software, Supervision, Funding acquisition, Validation, Investigation, Methodology, Writing – original draft, Project administration, Writing – review and editing; Chao Chen, Data curation, Software, Methodology, Writing – review and editing; Dong Wei, Qiang Hao, Resources, Investigation, Writing – review and editing; Yong Xu, Jianpo Zhai, Yaoguang Zhang, Pengjie Wu, Linlin Zhang, Resources, Writing – review and editing; Siying Liang, Resources, Investigation, Methodology, Writing – review and editing; Wenlong Jia, Resources, Software, Writing – review and editing; Jian Li, Data curation, Funding acquisition, Writing – review and editing; Yanchun Qu, Resources, Data curation, Writing – review and editing; Wei Zhang, Xinyu Yang, Data curation, Writing – review and editing; Lin Pan, Data curation, Methodology, Writing – review and editing; Ruomei Qi, Methodology, Writing – review and editing; Yao Li, Feiliang Wang, Rui Yi, Investigation,

Writing – review and editing; Ze Yang, Jianye Wang, Conceptualization, Supervision, Funding acquisition, Project administration, Writing – review and editing; Yanyang Zhao, Conceptualization, Data curation, Supervision, Funding acquisition, Investigation, Methodology, Writing – original draft, Project administration, Writing – review and editing

### Author ORCIDs
Xiaoquan Zhu ⓘ http://orcid.org/0000-0002-9099-5835
Yanyang Zhao ⓘ https://orcid.org/0000-0001-5139-5311

### Ethics
This study was conducted in line with the Declaration of Helsinki and the study protocol was approved by the Research Ethics Board of the Beijing Hospital, National Health Commission (2015BJYYEC-101-01). All participants provided written informed consent before participation.

This study was performed in accordance with the recommendations in the Guide for the Institutional Animal Care and Use Committee of National Institute of Biological Sciences. All of the animals were handled according to the protocol approved by the Committee of National Institute of Biological Sciences, Beijing (Beijing, China). All surgery was performed under sodium pentobarbital anesthesia to minimize suffering.

### Decision letter and Author response
Decision letter https://doi.org/10.7554/eLife.81258.sa1

## Additional files

### Supplementary files
• MDAR checklist

• Supplementary file 1. Details of the clinical information of the samples by RNA-seq, the fusion genes identified in this study, the gene set enrichment analyses conducted, the transgenic mouse prostates examined, and the primers used.

• Supplementary file 2. Correlation analyses in this study.

### Data availability
The RNA-Seq data from 10 pairs of primary prostate tumors and normal tissues has been deposited in GEO database with the accession number: GSE114740. The Whole Genome Sequencing data from FOXP2-CPED1 fusion-positive tumor (PC_1) has been deposited in SRA database with the accession number: SRR7223723.

The following datasets were generated:

| Author(s) | Year | Dataset title | Dataset URL | Database and Identifier |
|---|---|---|---|---|
| Zhu X, Zhao Y | 2021 | Expression profile of localized primary prostate cancer samples and the matched normal tissues | https://www.ncbi.nlm.nih.gov/geo/query/acc.cgi?acc=GSE114740 | NCBI Gene Expression Omnibus, GSE114740 |
| Zhu X, Zhao Y | 2019 | Whole genome sequencing of *Homo sapiens* : prostate | https://www.ncbi.nlm.nih.gov/sra/SRR7223723 | NCBI Sequence Read Archive, SRR7223723 |

The following previously published datasets were used:

| Author(s) | Year | Dataset title | Dataset URL | Database and Identifier |
|---|---|---|---|---|
| Moreno CS | 2014 | RNAseq Analysis of Formalin-Fixed Paraffin-Embedded Prostate Cancer Tissues | https://www.ncbi.nlm.nih.gov/geo/query/acc.cgi?acc=GSE54460 | NCBI Gene Expression Omnibus, GSE54460 |

*Continued on next page*

*Continued*

| Author(s) | Year | Dataset title | Dataset URL | Database and Identifier |
|---|---|---|---|---|
| Taylor BS, Schultz N, Hieronymus H, Sawyers CL | 2010 | Integrative genomic profiling of human prostate cancer | https://www.ncbi.nlm.nih.gov/geo/query/acc.cgi?acc=GSE21032 | NCBI Gene Expression Omnibus, GSE21032 |
| Armenia J | 2018 | Whole exome Sequencing of 1013 prostate cancer samples and their matched normals | http://www.cbioportal.org/study?id=prad_p1000 | cBioPortal, study?id=prad_p1000 |

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
