## [Editor Report]

The authors convincingly showed FOXP2 expression being associated with a high Gleason score, and ectopic expression of FOXP2 inducing malignant transformation of non-tumor cells. This is associated with increased MET signalling. With this, the authors position FOXP2 as bona-fide oncogene, driving prostate cancer development.

---

## [Decision Letter]

**Decision letter after peer review:**

Thank you for submitting your article "*FOXP2* confers oncogenic effects in prostate cancer through activating MET signalling" for consideration by *eLife*. Your article has been reviewed by 3 peer reviewers, one of whom is a member of our Board of Reviewing Editors, and the evaluation has been overseen by Paivi Ojala as the Senior Editor. The reviewers have opted to remain anonymous.

Essential revisions:

1) While the FOXP2 data are compelling and convincing, it is not clear yet whether this effect is specific, or if FOXP2 is e.g. universally relevant for cell viability. Targeting FOXP2 by siRNA/shRNA in a non-transformed cell line would address this issue.

2) No data are presented for how FOXP2 regulates MET signaling. While the authors provide correlative data suggesting that FOXP2 could increase the expression of MET signaling components, it is not clear how FOXP2 controls MET levels. It would be of interest to search for and validate the importance of potential FOXP2 binding sites in or around MET and the genes of MET-associated proteins. At a minimum, it should be confirmed whether MET is a primary or secondary target of FOXP2. ChIP would easily address if it is direct regulation of MET and analysis of FOXP2 ChIP-seq could provide insights.

3) The authors claim that FOXP2 functions as an oncogene, but the most-severe phenotype that is observed in vivo, is PIN lesions, not tumors. While this is an exciting observation, it is not the full story of an oncogene. Can the authors justifiably claim that FOXP2 is an oncogene, based on these results?

4) Beyond the 2 fusions in the 100 PCa patient cohort it is unclear how FOXP2 is overexpressed in PCa. In the discussion and in FS5 some data are presented indicating amplification and CNAs, however, these are not directly linked to FOXP2 expression.

5) There are some hints that full-length FOXP2 and the FOXP2-CPED1 function differently. In SF2E the size/number of colonies between full-length FOXP2 and fusion are different. If the assay was run for the same length of time, then it indicates different biologies of the over-expressed FOXP2 and FOXP2-CPED1 fusion. Additionally, in F3E the sensitization is different depending on the transgene.

6) The authors are requested to report on what happened to the 4-gene MET signature in the FOXP2 knockdown cell models.

7) The authors are requested to test if overexpression of MET can rescue the anti-growth effects of FOXP2 knockdown in prostate cancer cells (positive or negative results would be informative).

*Reviewer #1 (Recommendations for the authors):*

1. Full-size westerns are required to assess the quality of these data, and this should be provided in a revised version.

*Reviewer #2 (Recommendations for the authors):*

Increased FOXP2 expression in prostate cancer has been studied in depth in prior work such as that by Stumm et al. 2013 who evaluated FOXP2 protein levels via tissue microarrays covering >11,000 prostate cancer specimens. Moreover, FOXP2's functional pro-cancer role in prostate cancer has also been demonstrated by other groups. So, while the implication of FOXP2 as a novel oncogene and driver of prostate cancer initiation appears novel, its role in promoting prostate cancer is known. Hence, the claims that "the FOXP2 expression pattern in prostate cancer has remained unclear" should be corrected.

The authors suggest that FOXP2 expression could guide the use of MET-targeting agents. However, given the presented mechanism of action, it is unclear if this would be any better than simply assessing the expression of MET.

More can be added regarding the newly identified FOXP2-CPED1 fusion. The authors should include in the main figures a figure illustrating the newly identified FOXP2-CPED1 fusion compared to wildtype and the domains retained or lost. In addition, it would strengthen the study if the existence of the fusion is corroborated in other clinical cohorts and whether it tracks with disease severity or is it only found in indolent prostate cancers.

*Reviewer #3 (Recommendations for the authors):*

Other issues identified in the manuscript.

Identifying the fusion should not entirely be in the supplement. This is the initiating discovery that led to studying FOXP2 in PCa and explains the use of FOXP2-CPED1. Some panels in the supplement should be moved to the main manuscript.

The authors should be more explicit in their language when describing the soft agar experiments and make it very clear that many cell lines were made and tested in soft agar assays.

SF2J representative images are only high magnification and there is no quantification.

Figures 3A, 3B, 3F, and 3G should be quantified. These experiments should also be repeated (independent biological experiments) so that n=3. More details should also be provided. How long cells are treated with the drug for F3A, B is omitted. There are no total MET and AKT controls in F3C (critical to know if there is a stochiometric change in phosphorylation or if phospho-levels are up due to more protein). In F3I there is no vector control (do RWPE1 cells form colonies?). It is not clear how F3D and E are normalized. Are all samples normalized to the untreated vector, or is each line normalized to the untreated (is there an effect on 2D growth with FOXP2 over-expression)?

Figure 4C should be quantified. Higher magnification images (perhaps as an inset) should be shown. The pMET staining is unconvincing (perhaps too much counter-stain).

Not at all clear why FS5 data are in discussion. This data fits best in the Results.

Labeling of supplemental figures is not in accordance with *eLife*. There are some other formatting oddities, such as different font sizes in the manuscript.

[Editors' note: further revisions were suggested prior to acceptance, as described below.]

Thank you for resubmitting your work entitled "*FOXP2* confers oncogenic effects in prostate cancer through activating MET signalling" for further consideration by *eLife*. Your revised article has been evaluated by Paivi Ojala (Senior Editor) and a Reviewing Editor.

The manuscript has been improved but there are some remaining issues that need to be addressed, as outlined below:

Essential revisions:

Comment 7. The authors are requested to test if overexpression of MET can rescue the anti- growth effects of FOXP2 knockdown in prostate cancer cells (positive or negative results would be informative).

Answer 7. We thank the reviewer very much for the comment. We rescued the expression of HA-tagged MET in FOXP2 knockdown PC3 cells and performed a cell proliferation assay. We observed that MET restoration partially reversed the change in the viability of FOXP2-KD prostate cancer cells (Figure C in response letter), which further supports the involvement of MET in FOXP2-induced oncogenic effect.

Response: The data does not support the conclusion. First, statistics cannot be performed within one replicate, but needs to be done in comparing different biological replicates. In doing so, I would conclude there is no statistically significant difference when MET is overexpressed or not. The authors should either change this conclusion or provide sufficient convincing data that would support this conclusion (which is not the case now).

*Reviewer #1 (Recommendations for the authors):*

Comment 3. Unfortunately, not a single chemical inhibitor is truly 100% specific. Therefore, the Foretinib and MK2206 experiments should be confirmed using shRNAs/KOs targeting MEK and AKT. With the inclusion of such data, the authors would make a very compelling argument that indeed MEK/AKT signalling is driving the phenotype.

Answer 3. We thank the reviewer for highlighting this point and we agree with the reviewer's point that no chemical inhibitor is 100% specific. In this study, we used chemical inhibitors to provide further supportive data indicating that FOXP2 confers oncogenic effects by activating MET signaling. We characterized a FOXP2-binding fragment located in MET and HGF in LNCaP prostate cancer cells by utilizing the CUT&Tag method. We also found that MET restoration partially reversed oncogenic phenotypes in FOXP2-KD prostate cancer cells. All these data consistently supported that FOXP2 activates MET signaling in prostate cancer. Please refer to the "Answer to Essential Revisions #2 from the Editors" and to the "Answer to Essential Revisions #7 from the Editors" for details

Response: Without the use of shRNAs/siRNAs/KOs, the claim of MEK/AKT signalling driving the phenotype cannot be made. In this, the Cut& Tag experiments don't help to address the issue, and the requested knockdown/knockout experiments should be provided.

Comment 6. The clinical and phenotypic observations are exciting and relevant. The mechanistic insights of the study are quite limited in the current stage. How does FOXP2 give its phenotype, and result in increased MET phosphorylation? The association is there, but it is unclear how this happens.

Answer 6. We appreciate this valuable suggestion. In the current study, we used the CUT&Tag method to explore how FOXP2 activated MET signaling in LNCaP prostate cancer cells, and we identified potential FOXP2-binding fragments in MET and HGF. Therefore, we proposed that FOXP2 activates MET signaling in prostate cancer through its binding to MET and MET- associated gene. Please refer to the "Answer to Essential Revisions #2 from the Editors" for details.

Response: The Cut& Tag experiment provides information that MET may be under transcriptional control of FOXP2. However, this does not provide an indication of how it could affect signalling/ increased phosphorylation. This issue should be addressed

Comment 1. Full-size westerns are required to assess the quality of these data, and this should be provided in a revised version.

Answer 1. We thank the reviewer very much for the comment. As recommended, we provided all of the raw data from western blot analyses in the Source data files.

Response: Thank you for providing the raw data. However, in the current format, it's not possible for the reader to decipher what band is what. For example, for Figure 1 Supplement 2, raw data was provided, this is all is not labeled. The authors should provide proper labels to these files, so that it is clear which band in the raw data, was used in which subpanel of which figure.

*Reviewer #3 (Recommendations for the authors):*

Comment 4. Figure 4C should be quantified. Higher magnification images (perhaps as an inset) should be shown. The pMET staining is unconvincing (perhaps too much counter-stain).

Answer 4. We thank the reviewer very much for the valuable suggestions. We replaced the image with a clearer image and provided enlarged images in Figure 4C to make our data clearer.

Response: No quantification was provided. This should still be added.

[Editors' note: further revisions were suggested prior to acceptance, as described below.]

Thank you for resubmitting your work entitled "*FOXP2* confers oncogenic effects in prostate cancer through activating MET signalling" for further consideration by *eLife*. Your revised article has been evaluated by Paivi Ojala (Senior Editor) and a Reviewing Editor.

The manuscript has been improved but there are some remaining issues that need to be addressed, as outlined below:

Essential revisions:

Comment 7:

Original comment and response:

The authors are requested to test if overexpression of MET can rescue the anti- growth effects of FOXP2 knockdown in prostate cancer cells (positive or negative results would be informative).

Answer 7. We thank the reviewer very much for the comment. We rescued the expression of HA-tagged MET in FOXP2 knockdown PC3 cells and performed a cell proliferation assay. We observed that MET restoration partially reversed the change in the viability of FOXP2-KD prostate cancer cells (Figure C in response letter), which further supports the involvement of MET in FOXP2-induced oncogenic effect.

Response: The data does not support the conclusion. First, statistics cannot be performed within one replicate, but needs to be done in comparing different biological replicates. In doing so, I would conclude there is no statistically significant difference when MET is overexpressed or not. The authors should either change this conclusion or provide sufficient convincing data that would support this conclusion (which is not the case now).

Answer: We thank the editors for the comments very much. We have provided additional biological replicates then performed statistical analyses. Even though we observed that the expression of HA-tagged MET in FOXP2 knockdown PC3 cell clones partially reversed cell growth, given that the efficiency of stable expression of MET is relatively low, the difference in restoration did not reach statistical significance.

Further revisions requested following the third round of review:

Additional round of response: please specify how these results are implemented in the paper, as apparently the data do not support the original conclusion. These results should thus be incorporated in the paper, and the conclusion updated that MET restoration apparently does not reverse the change in viability of FOXP2-KD prostate cancer cells.*Reviewer #1:*

Comment 3:

Original comment and response:

Unfortunately, not a single chemical inhibitor is truly 100% specific. Therefore, the Foretinib and MK2206 experiments should be confirmed using shRNAs/KOs targeting MEK and AKT. With the inclusion of such data, the authors would make a very compelling argument that indeed MEK/AKT signalling is driving the phenotype.

Answer 3. We thank the reviewer for highlighting this point and we agree with the reviewer's point that no chemical inhibitor is 100% specific. In this study, we used chemical inhibitors to provide further supportive data indicating that FOXP2 confers oncogenic effects by activating MET signaling. We characterized a FOXP2-binding fragment located in MET and HGF in LNCaP prostate cancer cells by utilizing the CUT&Tag method. We also found that MET restoration partially reversed oncogenic phenotypes in FOXP2-KD prostate cancer cells. All these data consistently supported that FOXP2 activates MET signaling in prostate cancer. Please refer to the "Answer to Essential Revisions #2 from the Editors" and to the "Answer to Essential Revisions #7 from the Editors" for details

Response: Without the use of shRNAs/siRNAs/KOs, the claim of MEK/AKT signalling driving the phenotype cannot be made. In this, the Cut& Tag experiments don't help to address the issue, and the requested knockdown/knockout experiments should be provided.

Answer: Following the comment, we knocked down MET in human prostate epithelial cells RWPE-1 overexpressing FOXP2 or FOXP2-CPED1 and observed decreased MET expression and decreased phosphor-AKT in the cells. Subsequently, we observed that siRNA-mediated MET knockdown significantly decreased the growth of the RWPE-1 cells (Figure E)

Further revisions requested following the third round of review:

Additional response: this looks encouraging. Please specify how these conclusions and results are incorporated in the manuscript.

[Editors' note: further revisions were suggested prior to acceptance, as described below.]

Thank you for resubmitting your work entitled "*FOXP2* confers oncogenic effects in prostate cancer through activating MET signalling" for further consideration by *eLife*. Your revised article has been evaluated by Paivi Ojala (Senior Editor) and a Reviewing Editor.

The manuscript has been improved but there are some remaining issues that need to be addressed, as outlined below:

The authors now updated the text in the Results section, stating that "We also conducted an experiment to rescue the expression of HA-tagged MET in FOXP2 knockdown PC3 cells and performed a cell proliferation assay. Our observations showed that the expression of HA-tagged MET in the PC3 cell clones partially reversed cell growth. However, the difference in restoration did not reach statistical significance, suggesting that restoring MET does not reverse the change in viability of FOXP2 knockdown PC3 cells (Figure 3—figure supplement 1B)."

Their results show there is no difference (no significant difference = no difference), and the authors are requested to rephrase the first section of this paragraph, as the statement "Our observations showed that the expression of HA-tagged MET in the PC3 cell clones partially reversed cell growth. " is incorrect.

Along these lines, the new title of the manuscript "FOXP2 confers oncogenic effects in prostate cancer through activating MET signalling" is in disagreement with these results. No causal connection can be made based on these results, and the text, as well as the title of the manuscript, should reflect this. The authors need to change the title, so that it is in agreement with the results shown.

---

## [Author Response]

Essential revisions:1) While the FOXP2 data are compelling and convincing, it is not clear yet whether this effect is specific, or if FOXP2 is e.g. universally relevant for cell viability. Targeting FOXP2 by siRNA/shRNA in a non-transformed cell line would address this issue.

We appreciate the reviewer and editors for bringing to our attention the question of whether the effect of *FOXP2* is specific. As suggested, we knocked down *FOXP2* in human embryonic kidney 293 (HEK293) cells using two different siRNAs (#2, #3). HEK293 is considered a nontransformed cell line^1^, and the FOXP2 protein is present in relatively abundant quantities in the cells.

As shown in Author response image 1, immunoblot analysis revealed obviously decreased FOXP2 protein levels after knockdown by each of the two different siRNAs in the cells. The knockdown efficiency in HEK293 cells remained at >70% for 5 days after a single transfection of 50 nM siRNA. We found that siRNA-mediated *FOXP2* knockdown significantly decreased the growth of HEK293 cells.

**Author response image 1. sa2fig1:** A CellTiter-Glo assay was used to test the growth of HEK293 cells treated with the control siRNA and si*FOXP2* (#2 5’- TGGACAGTCTTCAGTTCTA-3’, #3 5’-CCACCAATAACTCATCATT-3’) over a fourpoint time course. Experiments were performed in triplicate and repeated five times with similar results. *Inset*, Protein blot showing knockdown of FOXP2 protein in cells. **P* < 0.05 and ***P* < 0.01 by the MannWhitney *U* test, mean ± SD; n = 5.

We had provided data related to the effects of *FOXP2* on NIH3T3 mouse primary fibroblasts (Figure 1J-1K and Figure1—figure supplement 3C-3D), which are a widely used cell model to demonstrate the presence of transformed oncogenes^2, 3^. When NIH3T3 cells acquired the exogenous *FOXP2* gene, we observed that the cells lost the characteristic contact inhibition response, continued to proliferate and eventually formed clonal colonies. As shown in Figure1—figure supplement 4D, frequent *FOXP2* gain was found in 18 different types of human tumors including prostate cancer, which may indicate the involvement of *FOXP2* in a broad spectrum of cancers. The latest research showed that *FOXP2* knockdown led to the decreased growth of lung cancer cells^4^. Together, these points suggested that the effects of *FOXP2* is not specifically relevant for prostate cell viability.

2) No data are presented for how FOXP2 regulates MET signaling. While the authors provide correlative data suggesting that FOXP2 could increase the expression of MET signaling components, it is not clear how FOXP2 controls MET levels. It would be of interest to search for and validate the importance of potential FOXP2 binding sites in or around MET and the genes of MET-associated proteins. At a minimum, it should be confirmed whether MET is a primary or secondary target of FOXP2. ChIP would easily address if it is direct regulation of MET and analysis of FOXP2 ChIP-seq could provide insights.

We appreciate the reviewer and editors for the comment, and we agree that it could be very informative, to test whether *MET* and its associated genes are bound by FOXP2 in prostate cancer. We employed Cleavage Under Targets and Tagmentation (CUT&Tag) strategy^5^, which is an alternative method of ChIP-seq. As shown in Figure 3J-3K and Figure 3figure supplement 1A-1B, we identified potential FOXP2-binding fragments located in *MET* and *HGF* in LNCaP prostate cancer cells. Combined with several lines of evidence supporting the implication of MET signaling in the *FOXP2*-induced oncogenic phenotype in vitro and in vivo in our study, our analysis of the CUT&Tag data further supported that FOXP2 activates MET signaling in prostate cancer.

We added the following sentence to the Results section:

“Additionally, to test whether *MET* and its associated genes are bound by FOXP2, we carried out Cleavage Under Targets and Tagmentation (CUT&Tag) assay (Kaya-Okur et al., 2019) in LNCaP cells overexpressing HAtagged *FOXP2*. We identified potential FOXP2-binding fragments located in *MET* and *HGF* in LNCaP prostate cancer cells (Figure 3J and K, Figure 3—figure supplement 1A-1B).”

3) The authors claim that FOXP2 functions as an oncogene, but the most-severe phenotype that is observed in vivo, is PIN lesions, not tumors. While this is an exciting observation, it is not the full story of an oncogene. Can the authors justifiably claim that FOXP2 is an oncogene, based on these results?

We thank the reviewer very much for the comment, and we agree with the concerns of the reviewer and editors. Although *FOXP2* induces malignant transformation of human prostate epithelial cells (RWPE-1) and can activate an oncogenic signaling pathway mediated by MET, we observed that *FOXP2* alone was not sufficient to cause cancer in the murine prostate. Therefore, it would be reasonable to propose that *FOXP2* has oncogenic potential. To make our conclusion more persuasive, we made the corresponding revision in the tracked changes version of the revised manuscript.

4) Beyond the 2 fusions in the 100 PCa patient cohort it is unclear how FOXP2 is overexpressed in PCa. In the discussion and in FS5 some data are presented indicating amplification and CNAs, however, these are not directly linked to FOXP2 expression.

We thank the reviewer and editors for the valuable suggestion. As recommended, we conducted the analysis on tumors with both *FOXP2* gene expression and amplification data available in the Catalogue Of Somatic Mutations In Cancer (COSMIC) datasets. In 268 tumors with *FOXP2* amplification at the DNA level, we observed consistent *FOXP2* mRNA overexpression across most tumors (197/268). These data suggested that *FOXP2* gain could be one of the possible reasons for the increased *FOXP2* gene expression in prostate cancer (Figure 1—figure supplement 4C). On the other hand, in 265 tumors with DNA methylation data available in COSMIC datasets, we found that *FOXP2* DNA methylation was consistently low in all 265 tumors, including 27 prostate tumors (Author response image 2), suggesting the epigenetic regulation of *FOXP2* expression in tumors.

**Author response image 2. sa2fig2:** Methylation status of *FOXP2* gene in tumors from COSMIC datasets.

We added the following sentence to the Discussion section:

“Moreover, we conducted the analysis on tumors with both *FOXP2* gene expression and amplification data available in the Catalogue Of Somatic Mutations In Cancer (COSMIC) datasets. In 268 tumors with *FOXP2* amplification at the DNA level, we observed consistent *FOXP2* mRNA overexpression across most tumors (197/268) (Figure 1—figure supplement 4C).”

5) There are some hints that full-length FOXP2 and the FOXP2-CPED1 function differently. In SF2E the size/number of colonies between full-length FOXP2 and fusion are different. If the assay was run for the same length of time, then it indicates different biologies of the over-expressed FOXP2 and FOXP2-CPED1 fusion. Additionally, in F3E the sensitization is different depending on the transgene.

We thank the reviewer very much for the comment. In our study, repeated four-week colony formation assays exhibited that 21 of the 42 NIH3T3 cell lines expressing the *FOXP2CPED*1 fusion gained the ability for anchorage-independent growth, as did 10 of the 32 cell lines with *FOXP2* expression, indicating that *FOXP2-CPED1* and *FOXP2* show comparable effects on cell transformation (*P* = 0.23, by Fisher’s exact test). Similar observations were in RWPE-1 human prostate epithelial cells (*P* = 0.69, by Fisher’s exact test). Our data from the xenograft model indicated that cells expressing *FOXP2* generated tumors in 10/15 NOD/SCID mice and that *FOXP2-CPED1* cells produced tumors in 11/15 mice. Moreover, *FOXP2* and the *FOXP2-CPED1* fusion can induce PIN in mice, but neither was sufficient to cause prostate cancer in the transgenic mouse model. Collectively, these data suggested that *FOXP2* and the *FOXP2-CPED1* fusion produced similar oncogenic phenotypes. We consider that the resemblance observed between *FOXP2* fusion- and wild-type *FOXP2*-overexpressing cells could be because the FOXP2 fusion product retained the intact zinc-finger motif and forkhead DNA binding domain (Ile502-Arg584) (Figure 1C), which are required for transcriptional regulation of target genes^6-9^. We added the corresponding description to the Results section.

6) The authors are requested to report on what happened to the 4-gene MET signature in the FOXP2 knockdown cell models.

We appreciate these helpful comments. As suggested, we tested the expression of the 4 genes (*HGF*, *MET*, *PIK3R1* and *PIK3CA*) in the *FOXP2* knockdown PC3 cells using qPCR. We found that the *HGF*, *MET* and *PIK3CA* were consistently downregulated in *FOXP2* knockdown PC3 cells.

We added the corresponding description to the Results section:

“Conversely, the core members (*HGF*, *MET*, and *PIK3CA*) were downregulated in the *FOXP2* knockdown PC3 prostate cancer cells (Figure 2—figure supplement 1F).”

7) The authors are requested to test if overexpression of MET can rescue the anti-growth effects of FOXP2 knockdown in prostate cancer cells (positive or negative results would be informative).

We thank the reviewer very much for the comment. We rescued the expression of HA-tagged MET in *FOXP2* knockdown PC3 cells and performed a cell proliferation assay. We observed that MET restoration partially reversed the change in the viability of *FOXP2*-KD prostate cancer cells (Author response image 3), which further support the involvement of MET in *FOXP2*-induced oncogenic effect.

**Author response image 3. sa2fig3:** Celltiter-Glo assay showed that HA-tagged MET overexpression was able to partially rescue the cell growth of PC3 prostate cancer cells stably expressing *FOXP2* shRNA (two clones, #21 and #23). *P* values calculated by the Mann-Whitney *U* test, mean ± s.d.; n = 4. Immunoblot showing ectopic expression of HA-tagged MET protein in PC3 cells stably expressing *FOXP2* shRNA.

Reviewer #1 (Recommendations for the authors):1. Full-size westerns are required to assess the quality of these data, and this should be provided in a revised version.

We thank the reviewer very much for the comment. As recommended, we provided all of the raw data from western blot analyses in the Source data files.

Reviewer #2 (Recommendations for the authors):1. Increased FOXP2 expression in prostate cancer has been studied in depth in prior work such as that by Stumm et al. 2013 who evaluated FOXP2 protein levels via tissue microarrays covering >11,000 prostate cancer specimens. Moreover, FOXP2's functional pro-cancer role in prostate cancer has also been demonstrated by other groups. So, while the implication of FOXP2 as a novel oncogene and driver of prostate cancer initiation appears novel, its role in promoting prostate cancer is known. Hence, the claims that "the FOXP2 expression pattern in prostate cancer has remained unclear" should be corrected.

We thank the reviewer for the correction. We agree with the reviewer’s concern. In the revised manuscript, to provide a more precise description, we deleted the sentence ‘the FOXP2 expression pattern in prostate cancer has remained unclear’ without changing the meaning.

2. The authors suggest that FOXP2 expression could guide the use of MET-targeting agents. However, given the presented mechanism of action, it is unclear if this would be any better than simply assessing the expression of MET.

We thank the reviewer for highlighting this point. In this study, we identified a significantly positive correlation between *FOXP2* expression and that of the four members of the MET pathway, including *HGF* and *MET*. Further functional assays showed that overexpression of *FOXP2* activated oncogenic MET signaling and that inhibition of MET signaling activation suppressed the *FOXP2*-induced oncogenic phenotype of human prostate cells. Moreover, we identified the binding of FOXP2 to *MET* and *HGF* in LNCaP prostate cancer cells. HGF binds its receptor MET, leading to MET activation. Therefore, we feel it would be worth evaluating *FOXP2* expression when the use of MET-targeting agents could be considered. We also agree that assessment of MET and phospho-MET protein level is an alternative strategy under such conditions.

3. More can be added regarding the newly identified FOXP2-CPED1 fusion. The authors should include in the main figures a figure illustrating the newly identified FOXP2-CPED1 fusion compared to wildtype and the domains retained or lost. In addition, it would strengthen the study if the existence of the fusion is corroborated in other clinical cohorts and whether it tracks with disease severity or is it only found in indolent prostate cancers.

We appreciate these valuable comments. As suggested, we moved a schematic view of the FOXP2-CPED1 fusion protein as well as a set of data regarding the newly identified *FOXP2* fusion to Figure 1 to make our manuscript more informative. In addition, we monitored the statuses of the two patients carrying the *FOXP2-CPED*1 fusion mutation (PC_1 and PC_11). At the 5-year follow-up, there was no recurrence and no indication of tumor metastasis after surgery, suggesting that the somatic fusion mutation might have occurred in indolent prostate tumors in the current study.

Reviewer #3 (Recommendations for the authors):Other issues identified in the manuscript.1. Identifying the fusion should not entirely be in the supplement. This is the initiating discovery that led to studying FOXP2 in PCa and explains the use of FOXP2-CPED1. Some panels in the supplement should be moved to the main manuscript.

We thank the reviewer for the comment. We have added a set of data regarding the newly identified *FOXP2* fusion in Figure 1 to make our manuscript more informative.

2. The authors should be more explicit in their language when describing the soft agar experiments and make it very clear that many cell lines were made and tested in soft agar assays.

We thank the reviewer very much for the correction. We added the corresponding information to the Results section:

“The assays showed that 21 of the 42 NIH3T3 cell lines expressing the *FOXP2-CPED*1 fusion gained the ability for anchorage-independent growth, as did 10 of the 32 cell lines with *FOXP2* expression, indicating that *FOXP2-CPED1* and *FOXP2* have comparable effects on cell transformation (*P* = 0.23, by Fisher’s exact test).”

3. Figures 3A, 3B, 3F, and 3G should be quantified. These experiments should also be repeated (independent biological experiments) so that n=3. More details should also be provided. How long cells are treated with the drug for F3A, B is omitted. There are no total MET and AKT controls in F3C (critical to know if there is a stochiometric change in phosphorylation or if phospho-levels are up due to more protein). In F3I there is no vector control (do RWPE1 cells form colonies?). It is not clear how F3D and E are normalized. Are all samples normalized to the untreated vector, or is each line normalized to the untreated (is there an effect on 2D growth with FOXP2 over-expression)?

We appreciate this valuable suggestion. Following this suggestion, we added the number of repetitions in the corresponding figure legends and performed quantitative analysis of the data from multiple western blot analyses. Please refer to corresponding Source data files for details. We also added detailed information regarding the duration of inhibitors in the Figure Legend of Figure 3F and 3G. We agree with the reviewer that total MET and AKT controls are critical in Figure 3C; however, not enough tumor material was available to perform the assessment. When RWPE-1 cells acquired the exogenous *FOXP2* or *FOXP2* fusion gene, we observed that the cells lost the characteristic contact inhibition response, continued to proliferate and eventually formed clonal colonies, but these effects were not observed in control RWPE-1 cells (Figure 1 L-1M and Figure1—figure supplement 3H). When calculating IC50 values, we normalized the viability rate of each cell line treated with different drug concentrations to that of the corresponding untreated cells (Figure 3D and 3E). We added the corresponding description to the tracked changes version of the revised manuscript.

4. Figure 4C should be quantified. Higher magnification images (perhaps as an inset) should be shown. The pMET staining is unconvincing (perhaps too much counter-stain).

We thank the reviewer very much for the valuable suggestions. We replaced the image with a clearer image and provided enlarged images in Figure 4C to make our data clearer.

5. Labeling of supplemental figures is not in accordance with eLife. There are some other formatting oddities, such as different font sizes in the manuscript.

We thank the reviewer very much for the correction. We carefully revised our manuscript and changed the layout in some figures in light of the corresponding comments.

Reference

Mayr C, Bartel DP. Widespread shortening of 3'UTRs by alternative cleavage and polyadenylation activates oncogenes in cancer cells. Cell. 2009 Aug 21;138(4):673-84.

[Editors' note: further revisions were suggested prior to acceptance, as described below.]

The manuscript has been improved but there are some remaining issues that need to be addressed, as outlined below:Essential revisions:Comment 7. The authors are requested to test if overexpression of MET can rescue the anti- growth effects of FOXP2 knockdown in prostate cancer cells (positive or negative results would be informative).Answer 7. We thank the reviewer very much for the comment. We rescued the expression of HA-tagged MET in FOXP2 knockdown PC3 cells and performed a cell proliferation assay. We observed that MET restoration partially reversed the change in the viability of FOXP2-KD prostate cancer cells (Figure C in response letter), which further supports the involvement of MET in FOXP2-induced oncogenic effect.Response: The data does not support the conclusion. First, statistics cannot be performed within one replicate, but needs to be done in comparing different biological replicates. In doing so, I would conclude there is no statistically significant difference when MET is overexpressed or not. The authors should either change this conclusion or provide sufficient convincing data that would support this conclusion (which is not the case now).

We thank the editors for the comments very much. We have provided additional biological replicates then performed statistical analyses. Even though we observed that the expression of HA-tagged MET in *FOXP2* knockdown PC3 cell clones partially reversed cell growth, given that the efficiency of stable expression of MET is relatively low, the difference in restoration did not reach statistical significance.

Comment 1. Full-size westerns are required to assess the quality of these data, and this should be provided in a revised version.Answer 1. We thank the reviewer very much for the comment. As recommended, we provided all of the raw data from western blot analyses in the Source data files.Response: Thank you for providing the raw data. However, in the current format, it's not possible for the reader to decipher what band is what. For example, for Figure 1 Supplement 2, raw data was provided, this is all is not labeled. The authors should provide proper labels to these files, so that it is clear which band in the raw data, was used in which subpanel of which figure.

Thank you for bringing our attention to data labeling. We followed the acquisition of the editorial support staff, as shown in the screenshot below, to upload two independent folders for each figure: one folder (folder name: Original files for figure) containing files of the raw, unedited blots without labels; one folder (folder name: Uncropped blot for figure) containing files of the uncropped gels with the relevant bands labeled. If it is preferable to provide labels in both types of files, we would be happy to do so.

Reviewer #3 (Recommendations for the authors):Comment 4. Figure 4C should be quantified. Higher magnification images (perhaps as an inset) should be shown. The pMET staining is unconvincing (perhaps too much counter-stain).Answer 4. We thank the reviewer very much for the valuable suggestions. We replaced the image with a clearer image and provided enlarged images in Figure 4C to make our data clearer.Response: No quantification was provided. This should still be added.

We have added the requested quantification analyses, as shown in Figure 4—figure supplement 1G, detailing the phosphorylation levels of Met and Akt in the mouse prostate in vivo. We added the corresponding description to Result section and Supplementary Figure Legends section.

Reference

Bottaro DP, Rubin JS, Faletto DL, Chan AM, Kmiecik TE, Vande Woude GF et al., Identification of the hepatocyte growth factor receptor as the c-met proto-oncogene product.

Science. 251(4995): 802-4.

2.Cooper CS. The met oncogene: from detection by transfection to transmembrane receptor for hepatocyte growth factor. Oncogene. 7(1): 3-7.3.Bardelli A, Longati P, Williams TA, Benvenuti S, Comoglio PM. A peptide representing the carboxyl-terminal tail of the met receptor inhibits kinase activity and invasive growth. J Biol Chem. 274(41):29274-81.4.Hov H, Holt RU, Rø TB, Fagerli UM, Hjorth-Hansen H, Baykov V et al., A selective c-met inhibitor blocks an autocrine hepatocyte growth factor growth loop in ANBL-6 cells and prevents migration and adhesion of myeloma cells. Clin Cancer Res. 10(19):6686-94.

[Editors' note: further revisions were suggested prior to acceptance, as described below.]

The manuscript has been improved but there are some remaining issues that need to be addressed, as outlined below:Essential revisions:Comment 7:Original comment and response:The authors are requested to test if overexpression of MET can rescue the anti- growth effects of FOXP2 knockdown in prostate cancer cells (positive or negative results would be informative).Answer 7. We thank the reviewer very much for the comment. We rescued the expression of HA-tagged MET in FOXP2 knockdown PC3 cells and performed a cell proliferation assay. We observed that MET restoration partially reversed the change in the viability of FOXP2-KD prostate cancer cells (Figure C in response letter), which further supports the involvement of MET in FOXP2-induced oncogenic effect.Response: The data does not support the conclusion. First, statistics cannot be performed within one replicate, but needs to be done in comparing different biological replicates. In doing so, I would conclude there is no statistically significant difference when MET is overexpressed or not. The authors should either change this conclusion or provide sufficient convincing data that would support this conclusion (which is not the case now).Answer: We thank the editors for the comments very much. We have provided additional biological replicates then performed statistical analyses. Even though we observed that the expression of HA-tagged MET in FOXP2 knockdown PC3 cell clones partially reversed cell growth, given that the efficiency of stable expression of MET is relatively low, the difference in restoration did not reach statistical significance.Further revisions requested following the third round of review:Additional round of response: please specify how these results are implemented in the paper, as apparently the data do not support the original conclusion. These results should thus be incorporated in the paper, and the conclusion updated that MET restoration apparently does not reverse the change in viability of FOXP2-KD prostate cancer cells.

We thank the editors very much for the helpful comments. In response to these suggestions, we have included the results in Figure 3—figure supplement 1 as panel B and added the corresponding description to the Results section:

“We also conducted an experiment to rescue the expression of HA-tagged MET in *FOXP2* knockdown PC3 cells and performed a cell proliferation assay. Our observations showed that the expression of HA-tagged MET in the PC3 cell clones partially reversed cell growth. However, the difference in restoration did not reach statistical significance, suggesting that restoring MET does not reverse the change in viability of *FOXP2* knockdown PC3 cells (Figure 3—figure supplement 1B).”

We have added the following sentences to the Figure Legend section:

“B. Celltiter-Glo assay was performed to test the effect of HA-tagged MET overexpression on the growth of PC3 prostate cancer cells stably expressing *FOXP2* shRNA (two clones, #21 and #23). The data of representative experiments resulted from three independent experiments with four replicates per group. *P* values calculated by 2-tailed Student’s t test, mean ± s.d.; n = 3. Immunoblot showing exogenous expression of HA-tagged MET protein in PC3 cells stably expressing *FOXP2* shRNA.”

We also added the corresponding description to the Materials and Method section and provided the corresponding source data.

Reviewer #1:Comment 3:Original comment and response:Unfortunately, not a single chemical inhibitor is truly 100% specific. Therefore, the Foretinib and MK2206 experiments should be confirmed using shRNAs/KOs targeting MEK and AKT. With the inclusion of such data, the authors would make a very compelling argument that indeed MEK/AKT signalling is driving the phenotype.Answer 3. We thank the reviewer for highlighting this point and we agree with the reviewer's point that no chemical inhibitor is 100% specific. In this study, we used chemical inhibitors to provide further supportive data indicating that FOXP2 confers oncogenic effects by activating MET signaling. We characterized a FOXP2-binding fragment located in MET and HGF in LNCaP prostate cancer cells by utilizing the CUT&Tag method. We also found that MET restoration partially reversed oncogenic phenotypes in FOXP2-KD prostate cancer cells. All these data consistently supported that FOXP2 activates MET signaling in prostate cancer. Please refer to the "Answer to Essential Revisions #2 from the Editors" and to the "Answer to Essential Revisions #7 from the Editors" for detailsResponse: Without the use of shRNAs/siRNAs/KOs, the claim of MEK/AKT signalling driving the phenotype cannot be made. In this, the Cut& Tag experiments don't help to address the issue, and the requested knockdown/knockout experiments should be provided.Answer: Following the comment, we knocked down MET in human prostate epithelial cells RWPE-1 overexpressing FOXP2 or FOXP2-CPED1 and observed decreased MET expression and decreased phosphor-AKT in the cells. Subsequently, we observed that siRNA-mediated MET knockdown significantly decreased the growth of the RWPE-1 cells (Figure E)Further revisions requested following the third round of review:Additional response: This looks encouraging. Please specify how these conclusions and results are incorporated in the manuscript.

We thank the editors very much for the valuable advice. We have added the results in Figure 3—figure supplement 1 as panel A and added the corresponding description to the Results section:

“Next, we knocked down *MET* expression in RWPE-1 cells that overexpressed the *FOXP2* or *FOXP2-CPED1*. We observed a decrease in both MET expression and phospho-AKT levels in these cells. Consistently, we found that siRNA-mediated knockdown of MET significantly reduced the growth of RWPE-1 cells (Figure 3—figure supplement 1A).”

We also added the following sentences to the Figure Legend section:

“A. Protein blot showing expression level of MET and AKT in RWPE-1 cells stably expressing *FOXP2*or*FOXP2-CPED1*treated with the control siRNA and si*MET*. Relative ratios of the intensities of the MET and p-AKT protein bands relative to the GAPDH band are shown bottom. CellTiter-Glo assay was performed to determine the growth of the corresponding RWPE-1cells treated with the control siRNA and si*MET,* respectively, over a three-point time course. The data resulted from three independent experiments with 4 replicates per group. *P* values calculated by 2-tailed Student’s *t*test, mean ± s.d.; n =3.”

We changed “inhibitors targeting MET signalling” to “inhibition of MET signalling” in the Abstract section, and we added the corresponding description to the Materials and Method section and provided the corresponding source data.

[Editors' note: further revisions were suggested prior to acceptance, as described below.]

The manuscript has been improved but there are some remaining issues that need to be addressed, as outlined below:The authors now updated the text in the Results section, stating that "We also conducted an experiment to rescue the expression of HA-tagged MET in FOXP2 knockdown PC3 cells and performed a cell proliferation assay. Our observations showed that the expression of HA-tagged MET in the PC3 cell clones partially reversed cell growth. However, the difference in restoration did not reach statistical significance, suggesting that restoring MET does not reverse the change in viability of FOXP2 knockdown PC3 cells (Figure 3—figure supplement 1B)."Their results show there is no difference (no significant difference = no difference), and the authors are requested to rephrase the first section of this paragraph, as the statement "Our observations showed that the expression of HA-tagged MET in the PC3 cell clones partially reversed cell growth. " is incorrect.Along these lines, the new title of the manuscript "FOXP2 confers oncogenic effects in prostate cancer through activating MET signalling" is in disagreement with these results. No causal connection can be made based on these results, and the text, as well as the title of the manuscript, should reflect this. The authors need to change the title, so that it is in agreement with the results shown.

We have changed the title to “*FOXP2* confers oncogenic effects in prostate cancer”.

We have rephased the description regarding the MET restoration in the Results section:

“We also conducted an experiment to rescue the expression of HA-tagged MET in *FOXP2* knockdown PC3 cells and performed a cell proliferation assay. However, the restoration of MET does not reverse the change in viability of *FOXP2* knockdown PC3 cells (Figure 3—figure supplement 1B).”

We removed a sentence from the Discussion section:

“Together, these data indicated that *FOXP2* is implicated in the initiation and progression of prostate cancer by aberrantly activating MET signalling.”